# Consensus Panel Recommendations for the Pharmacological Management of Pregnant Women with Depressive Disorders

**DOI:** 10.3390/ijerph20166565

**Published:** 2023-08-11

**Authors:** Georgios Eleftheriou, Riccardo Zandonella Callegher, Raffaella Butera, Marco De Santis, Anna Franca Cavaliere, Sarah Vecchio, Alessandra Pistelli, Giovanna Mangili, Emi Bondi, Lorenzo Somaini, Mariapina Gallo, Matteo Balestrieri, Umberto Albert

**Affiliations:** 1Italian Society of Toxicology (SITOX), via Giovanni Pascoli 3, 20129 Milan, Italy; 2Poison Control Center, Hospital Papa Giovanni XXIII, 24127 Bergamo, Italy; 3Italian Society of Psychiatry (SIP), piazza Santa Maria della Pietà 5, 00135 Rome, Italy; 4Psychiatry Unit, Department of Medicine (DAME), University of Udine, 33100 Udine, Italy; 5Italian Society of Obstetrics and Gynecology (SIGO), via di Porta Pinciana 6, 00187 Rome, Italy; 6Department of Obstetrics and Gynecology, Fondazione Policlinico Universitario A. Gemelli IRCCS, 00168 Rome, Italy; 7Department of Gynecology and Obstetrics, Fatebenefratelli Gemelli, Isola Tiberina, 00186 Rome, Italy; 8Ser.D Biella—Drug Addiction Service, 13875 Biella, Italy; 9Division of Clinic Toxicology, Azienda Ospedaliera Universitaria Careggi, 50134 Florence, Italy; 10Italian Society of Neonatology (SIN), Corso Venezia 8, 20121 Milan, Italy; 11Department of Neonatology, Hospital Papa Giovanni XXIII, 24127 Bergamo, Italy; 12Department of Psychiatry, ASST Papa Giovanni XXIII, 24100 Bergamo, Italy; 13Italian Society of Addiction Diseases (SIPAD), via Tagliamento 31, 00198 Rome, Italy; 14Italian Society for Drug Addiction (SITD), via Roma 22, 12100 Cuneo, Italy; 15Italian Society of Neuropsychopharmacology (SINPF), via Cernaia 35, 00158 Rome, Italy; 16Department of Medicine, Surgery and Health Sciences, University of Trieste, 34128 Trieste, Italy; 17Division of Clinic Psychiatry, Azienda Sanitaria Universitaria Giuliano-Isontina, 34148 Trieste, Italy

**Keywords:** pregnancy, antidepressants, anxiolytic drugs, consensus

## Abstract

Introduction: The initiative of a consensus on the topic of antidepressant and anxiolytic drug use in pregnancy is developing in an area of clinical uncertainty. Although many studies have been published in recent years, there is still a paucity of authoritative evidence-based indications useful for guiding the prescription of these drugs during pregnancy, and the data from the literature are complex and require expert judgment to draw clear conclusions. Methods: For the elaboration of the consensus, we have involved the scientific societies of the sector, namely, the Italian Society of Toxicology, the Italian Society of Neuropsychopharmacology, the Italian Society of Psychiatry, the Italian Society of Obstetrics and Gynecology, the Italian Society of Drug Addiction and the Italian Society of Addiction Pathology. An interdisciplinary team of experts from different medical specialties (toxicologists, pharmacologists, psychiatrists, gynecologists, neonatologists) was first established to identify the needs underlying the consensus. The team, in its definitive structure, includes all the representatives of the aforementioned scientific societies; the task of the team was the evaluation of the most accredited international literature as well as using the methodology of the “Nominal Group Technique” with the help of a systematic review of the literature and with various discussion meetings, to arrive at the drafting and final approval of the document. Results: The following five areas of investigation were identified: (1) The importance of management of anxiety and depressive disorders in pregnancy, identifying the risks associated with untreated maternal depression in pregnancy. (2) The assessment of the overall risk of malformations with the antidepressant and anxiolytic drugs used in pregnancy. (3) The evaluation of neonatal adaptation disorders in the offspring of pregnant antidepressant/anxiolytic-treated women. (4) The long-term outcome of infants’ cognitive development or behavior after in utero exposure to antidepressant/anxiolytic medicines. (5) The evaluation of pharmacological treatment of opioid-abusing pregnant women with depressive disorders. Conclusions: Considering the state of the art, it is therefore necessary in the first instance to frame the issue of pharmacological choices in pregnant women who need treatment with antidepressant and anxiolytic drugs on the basis of data currently available in the literature. Particular attention must be paid to the evaluation of the risk/benefit ratio, understood both in terms of therapeutic benefit with respect to the potential risks of the treatment on the pregnancy and on the fetal outcome, and of the comparative risk between the treatment and the absence of treatment; in the choice prescription, the specialist needs to be aware of both the potential risks of pharmacological treatment and the equally important risks of an untreated or undertreated disorder.

## 1. Introduction

The initiative of a consensus on the topic of antidepressant and anxiolytic drug use in pregnancy is developing in an area of clinical uncertainty. Although many studies have been published in recent years, there is still a paucity of authoritative evidence-based indications useful for guiding the prescription of these drugs during pregnancy; moreover, the data from the literature are complex and require expert judgment to draw clear conclusions. Antidepressant and anxiolytic drugs in pregnancy may constitute a potential risk to the developing fetus but are also a clear benefit for patients with acute symptoms or for those with a high risk of relapses or recurrences. On the other hand, clinicians, patients and their partners should always keep in mind that women with untreated depressive symptoms or major depressive disorder (MDD) may have several pregnancy complications (miscarriage, preterm birth, placental abruption, intrauterine hemorrhages) and may deliver newborns with an increased risk of preterm birth and small for gestational age status. Moreover, abrupt discontinuation of antidepressants during pregnancy may be associated with relapses/recurrences.

The art of medicine is exemplified by balancing the risks of untreated psychiatric disorders in general, depressive symptoms or MDD for example, during pregnancy and the risks of adverse outcomes associated with the exposure to medications (antidepressants in this case). There is the following common concern in prescribing medications, such as selective serotonin reuptake inhibitors (SSRIs), serotonin and norepinephrine reuptake inhibitors (SNRIs), tricyclic antidepressants (TCAs), atypical antidepressants, benzodiazepines (BDZs) and non–benzodiazepines hypnotics drugs (hypnotic benzodiazepine receptor agonists-HBRAs), during pregnancy: the possibility that prenatal psychopharmacologic treatment may prevent adverse outcomes during human pregnancy is less commonly considered than the possibility of fetal harm. This has led, at least in the past, to not prescribing medications to patients in need of them and/or at risk for suicide or self-harm or to immediately discontinue antidepressants for patients already on treatment, with potential adverse outcomes.

The objective of this consensus is to provide a concise and specific guide to the therapeutic choices to be implemented in pregnant women affected by depressive disorders.

## 2. Materials and Methods

For the elaboration of the consensus, it was considered appropriate to involve the scientific societies of the sector, namely, the Italian Society of Toxicology (SITOX), the Italian Society of Neuropsychopharmacology (SINPF), the Italian Society of Psychiatry (SIP), the Italian Society of Obstetrics and Gynecology (SIGO), the Italian Society of Drug Addiction (SITD) and the Italian Society of Addiction Pathology (SIPaD).

The chairperson and the coordinators of the consensus were members of the Italian Society of Toxicology and the Italian Society of Neuropsychopharmacology. They defined the objectives, scope, users and sections of the document, as well as the specific areas of the investigation. They selected a multidisciplinary group consisting of experts in research and management of anxiety and depressive disorders in pregnancy. They identified 8 leading Italian experts in women’s mental health, in neonatal complications and on the use of drugs in pregnancy and lactation. Selection of the expert group was based upon their practical input in diagnosis and treatment of major depression in pregnancy and their professional expertise in the field of the consensus. None of the experts had any potential conflicts of interest or funding sources in order to ensure the experts’ transparency and credibility. Our guidelines are not financially sponsored. All experts had long-term experience and reviewed publications in the field.

This interdisciplinary team was composed of experts from the following different medical specialties: 6 toxicologists and pharmacologists, 4 psychiatrists, 2 gynecologists, 1 neonatologist and included representatives of all the aforementioned scientific societies. The role of the toxicologists was the evaluation of the risk of malformations after in utero use of the medicines; the interest of the gynecologists and the neonatologist was the complications in pregnancy and the neonatal abstinence syndrome, respectively, and the psychiatrists evaluated the risks due to the psychiatric disease of the patients involved in all published articles.

Such team first identified the needs underlying the consensus using the methodology of the “Nominal Group Technique” (NGT). The NGT approach was chosen as it is a face-to-face group meeting process, generates a large number of ideas, allows for immediate feedback and is facilitated by experienced leaders. The NGT is a commonly used consensus method in medical research, which uses a panel of specialists to discuss and provide prompt results for researchers. NGT is highly structured and involves generation, clarification and sharing of ideas and voting.

In our study, from September 2022 to June 2023, several NGT meetings were conducted in order to reach a consensus statement on the pharmacological management of pregnant women with depressive disorders.

The first face-to-face NGT was held in Bergamo, 8–10 September 2022, during the 33rd Conference of the European Network of the Teratology Information Services, followed by adjustment of the first draft of consensus based on expert feedback via e-mail. Then, the team evaluated the most accredited international literature by a systematic review of the literature. Finally, after various online meetings, the research team reviewed all the materials and defined a list of consensus statements and through several discussion meetings, participants reached consensus on the draft and the final document.

The following six areas of investigation were identified (the order of presentation of the topics is intended to guide the clinical approach to women with a psychiatric disorder who intend to plan a pregnancy):Assessment of the risks associated with untreated maternal depression (depressive symptoms or MDD) during pregnancy;Assessment of the overall risk of malformations associated with antidepressant and anxiolytic drug use during pregnancy;Assessment of the risk of maternal complications during pregnancy and at delivery related to pharmacological treatment, such as gestational hypertension, pre-eclampsia, gestational diabetes, intrauterine growth retardation, spontaneous abortion, preterm birth and postpartum hemorrhage;Evaluation of neonatal adaptation disorders and persistent pulmonary hypertension of the newborn in the offspring of women exposed to antidepressants and/or anxiolytics during pregnancy;Long-term developmental outcomes of infants’ cognitive development or behavior after in utero exposure to antidepressant/anxiolytic medications;Evaluation of pharmacological treatment of opioid abuse pregnant women with depressive disorders.


**Definitions**


Major depressive disorder (MDD) is defined by the DSM–5 [1] as a common and potentially severe mood disorder (depressive disorder in DSM-5): diagnostic criteria require that the patient is experiencing, during the same 2-week period, five or more symptoms, at least one of which should be either (1) depressed mood or (2) markedly diminished interest or pleasure (see Box 1).Organ malformation (teratogenicity) was considered drug-induced if fetal drug exposure happened during the first 12 weeks of gestation when organ development occurs; in particular, a medication was considered teratogenic when prenatal exposure was associated with a significant increase in the risk of congenital physical deformities over the baseline risk. The baseline incidence of major congenital malformations (MM) ranges from 2% to 4% and has been estimated to be more than 15% if minor malformations are included [2,3,4].The following gestational and postpartum complications were considered as defined by the American College of Obstetricians and Gynecologists’ Committee on Practice Bulletins [5,6,7,8,9,10]: gestational hypertension, pre-eclampsia, gestational diabetes, intrauterine growth retardation, spontaneous abortion, preterm birth and postpartum hemorrhage (Box 2).Neonatal withdrawal syndrome was considered according to the definition provided by Anbalagan & Mendez [11]. The syndrome includes a wide range of physical and behavioral symptoms, usually associated with the use of drugs during pregnancy, appears shortly after birth and symptoms are typically of limited duration. Persistent pulmonary hypertension of the newborn (PPHN) is defined, according to Mandell et al., as the failure of the normal circulatory transition that occurs after birth. It is a syndrome characterized by marked pulmonary hypertension that causes hypoxemia secondary to right-to-left shunting of blood at the foramen ovale and ductus arteriosus [12].Developmental toxicity refers to the potential for long-term neurobehavioral abnormalities in children following in utero exposure to medications.Opioid use disorder is defined according to American Psychiatric Association DSM–5 criteria and includes a persistent desire or unsuccessful efforts to cut down or control opioid use despite negative social and/or professional consequences.

Box 1Symptoms of major depressive disorder (major depressive episode) (DSM-5).1.Depressed mood, most of the day, nearly every day.2.Markedly diminished interest or pleasure in all, or almost all, activities most of the day, nearly every day.3.Significant weight loss when not dieting or weight gain, or decrease or increase in appetite nearly every day.4.A slowing down of thought and a reduction of physical movement (observable by others, not merely subjective feelings of restlessness or being slowed down).5.Fatigue or loss of energy nearly every day.6.Feelings of worthlessness or excessive or inappropriate guilt nearly every day.7.Diminished ability to think or concentrate, or indecisiveness, nearly every day.8.Recurrent thoughts of death, recurrent suicidal ideation without a specific plan, or a suicide attempt or a specific plan for committing suicide.

Box 2Definition of common gestational and postpartum complications.*Gestational hypertension* is defined as systolic blood pressure ≥ 140 mmHg or diastolic blood pressure ≥ 90 mmHg, or both, on two occasions at least 4 h apart after 20 weeks of gestation, in woman with previously normal blood pressure.*Pre-eclampsia* is defined as the new-onset hypertension and proteinuria in pregnancy or new-onset hypertension with the new onset of any of the following: thrombocytopenia with platelets < 100 × 10^9^/L or renal insufficiency with serum creatinine > 1.1 mg/dL or impaired liver function with elevated blood concentrations of liver transaminases to twice normal concentration or pulmonary edema or unexplained new-onset headache unresponsive to medication—without an alternative diagnosis—or visual symptoms.*Gestational diabetes mellitus* is defined as any degree of glucose intolerance with onset or first recognition during pregnancy.*Postpartum hemorrhage* is defined as cumulative blood loss of greater than or equal to 1000 mL or blood loss accompanied by signs or symptoms of hypovolemia within 24 h after the birth.*Intrauterine growth retardation* describes fetuses with an estimated fetal weight or abdominal circumference that is less than the 10th percentile for gestational age.*Preterm birth* is defined as a delivery occurring at or after 20 (+0/7 d) weeks of gestation and before 37 (+0/7 d) weeks of gestation.*Spontaneous abortion or early pregnancy loss* is defined as a nonviable, intrauterine pregnancy with either an empty gestational sac or a gestational sac containing an embryo or fetus without fetal heart activity within the first 12 weeks of gestation.

### Search Strategy

A systematic review of the literature was performed to assess the risks associated with untreated maternal depression in pregnancy and the risk of malformations with the antidepressant and anxiolytic drugs used in pregnancy. The search strategies for the PubMed, EMBASE and Cochrane library databases were defined using a list of combined generic keywords and medical subject headings. In order to be included, studies had to fulfill the following criteria: (1) they had to include pregnant patients with depression; (2) they had to provide data on pregnancy and delivery and on drug exposure during pregnancy; and (3) they had to be a meta-analysis, systematic review of the literature or observational studies.

The search for relevant studies using generic keywords and MeSH terms was the following: “depression during pregnancy” AND “malformations” OR “teratogenicity” OR “pregnancy outcomes” OR “perinatal outcome” OR “neonatal outcome” AND “selective serotonin reuptake inhibitors” OR “serotonin and norepinephrine reuptake inhibitors” OR “tricyclic antidepressants” OR “atypical antidepressants” OR ”benzodiazepines” OR “hypnotic benzodiazepine receptor agonists” OR “opioids” AND “during pregnancy”. Namely, the following compounds were considered: citalopram, escitalopram, fluoxetine, fluvoxamine, paroxetine and sertraline for the SSRIs; venlafaxine and duloxetine for the SNRIs; vortioxetine for the Serotonin Modulator and Stimulator antidepressants (SMS); bupropion, mirtazapine, reboxetine and trazodone for the atypical antidepressants; amitriptyline and clomipramine for the TCAs.

Periods between 1 January 1995 and 1 January 2023 were addressed. This information enabled the group to develop a series of recommendations.

## 3. Results

We have identified 620 records about the use of antidepressants during pregnancy, and after the exclusion of other languages other than English and experimental studies, we found 417 eligible studies. Searching only the full-text articles, we found 398 records. From them, we excluded 61 papers concerning case reports and case series, and 43 records were excluded for other reasons. From the remaining records, 30 reports were not retrieved. The research strategy allowed the retrieval of 264 references, and a total of 6 recommendations were generated. The literature results about the risk of congenital malformations are summarized in tables below.

The group of SSRIs accounted in 17 meta-analyses and 51 cohort and case–control studies (32 retrospective, 17 prospective and 2 population-based cohort studies); the SNRIs one was composed of 2 meta-analyses, 8 retrospective, 3 prospective and 2 population-based cohort studies. The tricyclic antidepressant and the atypical antidepressant group were composed of 1 meta-analysis, 7 retrospective, 1 prospective and 1 population-based cohort studies and 1 meta-analysis, 5 retrospective, 5 prospective and 1 population-based cohort studies, respectively. Eventually, the BDZs and HBRAs accounted in 4 meta-analyses, 9 retrospective and 2 prospective cohort studies.

### 3.1. Major Malformations and Cardiac Malformations

#### 3.1.1. Meta-Analyses

We found 17 meta-analyses about the risk of MM or cardiac malformations (CM) after exposure to SSRI (Table 1). The vast majority of them did not control for possible confounding by indication as follows: 8 meta-analyses evaluated MM risk, 5 of which resulted in significant risk [13,14,15,16,17] and 3 in non-significant risk [18,19,20]; 18 meta-analyses evaluated CM risk, 11 of which report significant risk [13,16,17,19,21,22,23,24,25,26,27], while 7 report non-significant CM risk [14,15,20,23,24,28,29]. Looking for meta-analyses that accounted for possible confounding by indication, we found only two studies [17,24], both of which resulted in a non-significant risk for congenital malformations overall. We found two meta-analyses that evaluated malformation risk after SNRI antidepressant exposure during pregnancy (Table 2), which reported significant CM risk [27,30] but non-significant MM risk [30]. Only one evaluated possible confounding by treatment indication [30], resulting in non-significant MM and CM risk. We also found 1 meta-analysis that evaluated malformation risk for exposure during pregnancy to TCA (Table 3), which concluded for non-significant CM risk, and to bupropion (Table 5), it reported significant CM risk [27]. This meta-analysis did not control for possible indication bias.

*Key findings:* MM and CM aggregate risk results are contradictory. When the underlying psychiatric condition is considered, aggregate MM or CM risk results as non-significant.

**Table 1 ijerph-20-06565-t001:** SSRI use in pregnancy and risk of malformations: meta-analyses.

Study	Design of the Study	Drug Class/Drug	Main Results (vs Not Exposed, General Population)	Main Results (vs Not Exposed, Psychiatric Population)
Einarson et al., 2005 [18]	Meta-analysis	Antidepressants	Non-significant MM risk (RR = 1.01; 95% CI 0.57–1.80)	Not controlled
Rahimi et al., 2006 [19]	Meta-analysis	SSRI	Non-significant MM risk (OR = 1.39; 95% CI 0.91–2.15), non-significant CM risk (OR = 1.19; 95% CI 0.53–2.68)	Not controlled
Bar-Oz et al., 2007 [13]	Meta-analysis	Paroxetine	Significant risk for MM (OR = 1.31; 95%CI 1.03–1.67) Significant risk for CM (OR = 1.72; 95% CI 1.22–2.42) Controlling vs. other AD: non-significant MM risk (OR = 1.30; 95% CI 0.93–1.80), non-significant CM risk (OR = 1.70; 95% CI 1.17–2.46)	Not controlled
O’Brien et al., 2008 [28]	Meta-analysis	Paroxetine	Non-significant risk for CM (OR = 1.18; 95% CI 0.88–1.59)	Not controlled
Nikfar et al., 2012 [14]	Meta-analysis	SSRI	Significant risk for MM (OR = 1.27; 95% CI 1.09–1.47); Non-significant risk for CM (OR = 1.19; 95% CI 0.39–3.64)	Not controlled
Grigoriadis et al., 2013 [21]	Meta-analysis	Antidepressants	Non-significant risk MM (RR = 1.07; 95% CI, 0.99–1.17) Significant risk for CM (RR = 1.36; 95% CI, 1.08–1.71) and for Septal Heart Defects (RR = 1.40; 95% CI, 1.10–1.77). Significant pooled effect for paroxetine (RR = 1.43; 95% CI, 1.08–1.88). Non-significant effect for fluoxetine.	Only one study controlled for psychiatric disorder [31]
Myles et al., 2013 [15]	Meta-analysis	SSRI	Significant MM risk (OR = 1.10; 95% CI 1.03–1.16). Non-significant CM risk (OR = 1.15; 95% CI 0.99−1.32). Paroxetine: significant risk for MM (OR = 1.29; 95% CI 1.11–1.49) and CM (OR = 1.44; 95% CI 1.12–1.86). Fluoxetine: significant risk for MM (OR = 1.14; 95% CI 1.01–1.30). Non-significant risk for CM (OR = 1.25; 95% CI 0.98–1.60). Non-significant risk for MM and CM for citalopram and sertraline.	Not controlled
Painuly et al., 2013 [22]	Meta-analysis	Paroxetine	Significant risk for CM (pooled RR = 1.25; 95% CI 1.01–1.54) Non-significant risk for CM in case–control studies (RR = 1.09; 95% CI 0.91–1.30). Non-significant risk for CM in cohort studies (RR = 1.52; 95% CI 0.98–2.34)	Not controlled
Riggin et al., 2013 [23]	Meta-analysis	Fluoxetine vs non-exposed to SSRI	Non-significant risk for MM (OR = 1.12; 95% CI 0.98–1.28) Significant risk for CM in cohort studies (OR = 1.6; 95% CI 1.31–1.95), non-significant risk for CM in case–control studies (OR = 0.63; 95% CI 0.39–1.03).	Not controlled
Bérard et al., 2016 [16]	Meta-analysis	Paroxetine	Significant risk for MM (OR = 1.23; 95% CI 1.10–1.38) Significant risk for CM (OR = 1.28; 95% CI 1.11–1.47)	Not controlled One study [32] controlled for depression: non-significant risk for MM (Adjusted OR = 1.08; 95% CI 0.77–1.50)
Wang et al., 2015 [29]	Meta-analysis	SSRI	Non-significant risk for CM (pooled OR = 1.06; 95% CI 0.94–1.18). Non-significant risk stratifying for singular AD.	Not controlled
Jordan et al., 2016 [24]	Meta-analysis	SSRI	Significant risk for composite adverse outcome of ’anomaly or stillbirth’ (OR = 1.13; 95% CI 1.03–1.24), dose–response relationship (OR = 1.10; 95% CI 1.02–1.20). Non-significant any M risk (aOR = 1.08; 95% CI 0.97–1.20). Non-significant CM risk (aOR= 1.00; 95% CI 0.82–1.21). Significant risk for severe CM (OR = 1.50; 95% CI 1.06–2.11), dose–response relationship (meta-regression OR = 1.49; 95% CI 1.13–1.97) Limited controlling for drug co-exposure.	Controlled for depression (lifetime): Non-significant risk for “anomaly or stillbirth” (OR= 1.00; 95% CI 0.82–1.26); non-significant any M risk (OR = 1.00; 95% CI 0.79–1.27); Non-significant CM risk (OR = 0.69; 95% CI 0.44–1.08), non-significant severe CM risk (OR = 1.46; 95% CI 0.61–3.53).
Kang et al., 2017 [20]	Meta-analysis	Citalopram	Non-significant risk for MM (OR = 1.07; 95% CI 0.98–1.17). Non-significant risk for CM (OR = 1.31; 95% CI 0.88–1.93) (after one outlier exclusion OR = 1.03; 95% CI 0.84–1.26)	Not controlled
Shen et al., 2017 [25]	Meta-analysis	Sertraline	Non-significant risk for MM 1.14 (0.66–2.13); Significant risk for CM (OR = 1.36; 95% CI 1.06–1.74)	Not controlled
Gao et al., 2018 [17]	Meta-analysis	SSRI	Significant risk for MM (RR = 1.11; 95% CI 1.03–1.19); Significant risk for CM (RR = 1.24; 95% CI 1.11–1.37)	Controlled for psychiatric diagnosis (mainly depression and anxiety): Non-significant risk for MM (RR = 1.04; 95% CI 0.95–1.13) Non-significant risk for CM (RR = 1.06; 95% CI 0.90–1.26) Similar results restricting to every single SSRI
Biffi et al., 2020 [26]	Umbrella review of 22 meta-analyses	SSRI	Significant risk for cardiovascular M (RR = 1.26; 95% CI 1.13–1.39) Significant risk for CM (RR = 1.17; 95% CI 1.06–1.28)	Not controlled
De Vries et al., 2021 [27]	Meta-analysis	SSRI (I trimester exposure)	Any AD significant CM risk (OR = 1.28; 95% CI 1.17–1.41) SSRI significant CM risk (OR = 1.25; 95% CI 1.15–1.37)	Not controlled

MM = major malformations; CM = cardiac malformations; SSRI = selective serotonin reuptake inhibitors; SNRI= serotonin-noradrenaline reuptake inhibitors; AD = antidepressant; OR = odd ratio; aOR = adjusted OR; RR = relative risk; aRR = adjusted RR; CI = confidence interval.

#### 3.1.2. Cohort and Case–Control Studies

We looked for cohort and case–control studies evaluating MM risk and CM risk after exposure to antidepressants in pregnancy. We found 51 studies for SSRI antidepressants (Table 2). The vast majority of studies did not control for possible confounding by indication as follows: 25 studies evaluated MM risk, 5 resulted in significant risk [33,34,35,36,37] and 20 resulted in non-significant MM risk [32,33,34,38,39,40,41,42,43,44,45,46,47,48,49,50,51,52,53,54]; 25 studies evaluated the cardiac malformations risk, 10 of these resulted in significant CM risk [35,36,37,50,55,56,57,58,59,60] and 15 resulted in non-significant CM risk [32,34,43,44,45,48,51,53,54,57,61,62,63,64]. After controlling for indication, or at least partially accounting for an underlying psychiatric condition, no study concluded for significant MM risk, while 10 resulted in non-significant MM risk [31,32,36,62,64,65,66,67,68,69]; 9 studies controlled for an indication for CM risk, 1 resulted in significant CM risk [64] while 8 resulted in non-significant CM risk [31,32,36,56,59,62,64,68].

We found the following 11 studies evaluating malformation risk for specific SSRIs (paroxetine, fluoxetine and citalopram): 1 study was significant for MM risk [70] while 5 resulted non-significant for MM risk [71,72,73,74,75]; one study reported significant CM risk [76] and 5 non-significant CM risk [70,76,77,78,79]. No study evaluated a single SSRI controlling for possible indication bias. Our research found 15 studies that evaluated malformation risk after exposure to SNRI antidepressants during pregnancy (Table 3). Most of these studies did not control for possible indication bias as follows: nine studies evaluated MM risk; three studies resulted in significant risk [59,80,81], while six resulted in non-significant MM risk [46,49,82,83,84,85]. Six studies evaluated CM risk; three studies resulted in significant risk [56,59,80], while three resulted in non-significant CM risk [37,63,64]. A total of seven studies examined malformation risk after SNRI exposure controlling for psychiatric symptoms as follows: one study resulted in significant MM risk [59] while four resulted in non-significant MM risk [67,81,85,86]; one study resulted in significant CM risk [59] while three resulted in non-significant CM risk [56,65,86].

We found 10 studies that evaluated malformation risk after exposure to TCA antidepressants during pregnancy (Table 4). Most of these studies did not control for possible indication bias as follows: three studies evaluated MM risk; one resulted in significant risk [49] while two resulted in non-significant MM risk [32,44]; six studies evaluated CM risk, three resulted in significant risk [41,49,64] while three resulted in non-significant CM risk [32,44,56]. After controlling for indication, or at least partially accounting for an underlying psychiatric condition, no study resulted in a significant risk of MM, while four resulted in non-significant MM risk [31,32,65,69], only one resulted in significant risk for CM [64] while three resulted in non-significant CM risk [32,56,65].

We also found 12 studies that evaluated malformation risk after exposure to multimodal antidepressants during pregnancy (Table 5). Of these, the majority reported results not controlled for possible indication bias as follows: seven studies evaluated MM risk; one resulted in significant risk [85] while six resulted in non-significant MM risk [49,87,88,89,90,91]; five studies evaluated CM risk, one resulted in significant risk [64] while four resulted in non-significant CM risk [49,56,59,92]. After controlling for indication, or at least partially accounting for an underlying psychiatric condition, no study resulted in significant risk for MM, while two resulted in non-significant MM risk [85,93]; only one study reported significant CM risk [64] while three studies resulted in non-significant CM risk [56,59,93].

We found the following four meta-analyses that summarized malformation risk after exposure to BDZ during pregnancy: one [94], not controlling for possible confounding by underlying psychiatric condition, reported significant pooled MM risk for case–control studies but non-significant pooled risk for cohort studies; two meta-analyses [95,96] reported non-significant MM or CM risk overall, even after controlling for possible indication bias [96]. We found 11 studies that evaluated malformation risk after exposure to BDZ or HBRA during pregnancy (Table 6). Of these, six reported data not controlled for possible indication bias, eight studies evaluated MM risk, one study resulted in significant risk [97] while seven resulted in non-significant MM risk [53,97,98,99,100,101,102]; four studies evaluated CM risk, no study resulted in significant risk while all of them resulted in non-significant CM risk [32,53,99,101]. After controlling for indication, or at least partially accounting for an underlying psychiatric condition, one study resulted in significant risk [103] while five resulted in non-significant MM risk [62,98,100,102,104]; one study resulted in significant risk for CM [103] while two resulted in non-significant CM risk [62,102].

*Key findings:* MM and CM risk results tended toward overall non-significant risk (124 vs. 37 results). When an underlying psychiatric condition is considered, MM or CM risk results as non-significant.

**Table 2 ijerph-20-06565-t002:** SSRI use in pregnancy and risk of malformations: prospective and retrospective cohort and case–control studies.

Study	Design of the Study	Drug Class/Drug	Main Results (vs Not Exposed, General Population)	Main Results (vs Not Exposed, Psychiatric Population)
Kulin et al., 1998 [38]	Prospective controlled cohort study	SSRI	Non-significant MM risk (RR = 1.06; 95% CI 0.43–2.62)	Not controlled
Ericson et al., 1999 [39]	Prospective cohort study	SSRI, TCAs, other	Non-significant congenital malformation risk	Not controlled
Simon et al., 2002 [65]	Retrospective cohort study	SSRI		Controlled for depression history (not clear if during pregnancy): Non-significant risk for MM (OR = 1.36; 95% CI 0.56–3.30)
Hendrick et al., 2003 [40]	Prospective observational study	SSRI	Rate of congenital malformation comparable to general population	Not controlled
Källén et al., 2003 [41]	Retrospective case–control study	Antidepressants (SSRI, TCA)	Non-significant CM risk (OR = 1.14; 95% CI 0.83–1.56) Non-significant CM risk for SSRI (OR = 0.95; 95% CI 0.62–1.44)	Not controlled
Malm et al., 2005 [42]	Retrospective register-based cohort study	SSRI	Non-significant risk for MM (aOR = 1.0; 95% CI 0.6–1.7) (fluvoxamine not analyzed)	Not controlled
Wen et al., 2006 [43]	Retrospective cohort study	SSRI	Non-significant risk for MM (including CM) (aOR = 0.98; 95% CI 0.59–1.64)	Not controlled
Wogelius et al., 2006 [33]	Retrospective register-based cohort study	SSRI (early exposure)	Non-significant risk for congenital M (aRR = 1.34; 95% CI 1.00–1.79) Significant risk for II-III trimester exposure for congenital M (aRR = 1.84; 95% CI 1.25–2.71) but no statistical evaluation of the difference with early exposure.	Not controlled
Alwan et al., 2007 [34]	Retrospective case–control study	SSRI vs. non-SSRI exposure	Paroxetine: Non-significant risk for MM (OR = 1.6; 95% CI 0.9–2.7). Non-significant risk for CM (OR = 1.6; 95% CI 0.9–2.7). Non-significant risk for MM or CM for fluoxetine, sertraline and citalopram.	Not controlled
Bérard et al., 2007 [66]	Record-linkage retrospective case–control study	SSRI vs. non-SSRI (I trimester exposure)	Non-significant risk for MM (OR = 1.23; 95% CI 0.88–1.70)	Controlling for depression during 1 year before or during pregnancy (between many other confounders): Other-than-paroxetine SSRI: non-significant MM risk (aOR = 0.93; 95% CI 0.53–1.62), non-significant CM (aOR = 0.89; 95% CI 0.28–2.84). Paroxetine: non-significant MM risk (aOR = 1.32; 95% CI 0.79–2.20), non-significant CM (aOR = 1.38; 95% CI 0.49–3.92), Paroxetine > 25 mg/day: significant MM risk (aOR = 2.23; 95% CI 1.19–4.17), non-significant CM risk (aOR = 3.07; 95% CI 1.00–9.42)
Davis et al., 2007 [44]	Register-based retrospective cohort study	SSRI (I trimester exposure)	Non-significant risk for MM (RR = 0.97; 95% CI 0.81–1.16). Non-significant risk for CM (RR = 0.93; 95% CI 0.50–1.73)	Not controlled
Källén et al., 2007 [45]	Retrospective population-based cohort study	SSRI (I trimester exposure)	Non-significant risk for congenital M (aOR = 0.89; 95% CI 0.79–1.07); non-significant CM risk (OR = 0.97; 95% CI 0.77–1.21) Significant risk for CM for paroxetine after adjustment for previous miscarriage (aOR = 1.63; 95% CI 1.05–2.53) and for teratogenic drug exposure (aRR = 2.93; 95% CI 1.52–5.13)	Not controlled
Lennestål et al., 2007 [46]	Retrospective cohort study	SSRI	Non-significant congenital M risk (OR = 0.89; 95% CI 0.79–1.07)	Not controlled
Louik et al., 2007 [61]	Retrospective case–control study	SSRI	Non-significant risk for MM (OR = 0.89; 95% CI 0.73–1.00). Non-significant risk for CM (aOR = 1.2; 95% CI 0.90–1.60), significant right ventricular outflow tract obstruction risk (OR = 2.0; 95% CI 1.1–3.6)	Not controlled
Oberlander et al., 2008 [62]	Retrospective register-based cohort study	SSRI	Non-significant risk for MM (OR = 0.91; 95% CI 0.72–1.14) Non-significant risk for CM (OR = 0.65; 95% CI 0.40–1.03)	Controlled for any ICD-9 diagnosis containing depression (and dosage and duration of exposure): non-significant MM risk difference (adjusted RD = −0.61; 95% CI −1.44–0.21); non-significant CM, (aRD = 0.21; 95% CI −0.14–0.56). Significant atrial septal defects. Non-significant MM for single SSRI, significant CM for citalopram (aRD = 2.28; 95% CI 0.19–4.36)
Ramos et al., 2008 [31]	Retrospective case–control study	SSRI		Controlled for depression/anxiety (during pregnancy): non-significant first-trimester MM risk for non-paroxetine-SSRI (aOR = 1.19; 95% CI 0.71–1.97), non-significant MM risk for paroxetine (aOR = 1.27; 95% CI 0.78–2.06). Non-significant duration-of-exposure MM risk (any AD)
Einarson et al., 2009 [47]	Prospective controlled cohort study	Antidepressants (I trimester exposure)	Non-significant risk for MM for AD (RR = 0.96; 95% CI 0.55–1.67). Non-significant risk for single AD.	Not controlled
Merlob et al., 2009 [55]	Prospective cohort study	SSRI (+ venlafaxine)	Significant risk for mild CM (RR = 2.17; 95% CI 1.07–4.39). (no CM attributable to venlafaxine)	Not controlled
Pedersen et al., 2009 [48]	Retrospective population-based cohort study	SSRI	Non-significant risk for minor M M (aOR = 0.88; 95% CI 0.54–1.41), non-significant MM risk (aOR = 1.21; 95% CI 0.91–1.62) Non-significant risk for CM (aOR = 1.44; 95% CI 0.86–2.40). Significant risk for septal heart defects (aOR = 1.99; 95% CI 1.13–3.53) (fluvoxamine not analyzed)	Not controlled
Wichman et al., 2009 [63]	Retrospective cohort study	SSRI	Non-significant risk for CM (OR = 0.44; 95% CI 0.14–1.38) Non-significantly different congenital heart disease rate (*p* = 0.23)	Not controlled
Kornum et al., 2010 [35]	Prospective population-based study	SSRI (early exposure)	Significant risk for congenital M (OR = 1.3; 95% CI 1.1–1.6) Significant risk for CM (OR = 1.7; 95% CI 1.1–2.5), significant risk for CM for sertraline (OR = 3.0; 95% CI 1.4–6.4).	Not controlled
Reis et al., 2010 [49]	Retrospective cohort study	SSRI	Significant risk for MM (OR = 1.20; 95% 1.10–1.31). Non-significant MM risk (aOR = 1.08; 95% CI 0.97–1.21) Non-significant cardiovascular M (aOR = 0.99; 95% CI 0.82–120); significant risk for cardiovascular M to paroxetine (aOR = 1.81; 95% CI 1.19–2.76)	Not controlled
Colvin et al., 2011 [50]	Retrospective population-based study	SSRI	Non-significant risk for MM (OR = 1.05; 95% CI 0.87–1.27). Significant risk for CM (OR = 1.60; 95% CI 1.10–2.31).	Not controlled
Malm et al., 2011 [51]	Retrospective register-based cohort study	SSRI	Non-significant risk MM (aOR = 1.08; 95% CI 0.96–1.22). Non-significant risk CM (aOR = 1.09; 95% CI 0.90–1.32), significant CM risk for fluoxetine (aOR = 1.40; 95% CI 1.01–1.95)	Not controlled
Colvin et al., 2012 [52]	Retrospective cohort study	SSRI	Non-significant any major birth defect risk (OR = 1.05; 95% CI 0.45–2.46)	Not controlled
Jimenez-Solem et al., 2012 [36]	Register-based retrospective cohort study	SSRI	Significant risk for MM (OR = 1.33; 95% CI 1.16–1.53) Significant risk for CM (aOR = 2.01; 95% CI 1.60–2.53)	First trimester exposure OR compared to no exposure between 3 months before conception and 1 month after giving birth OR: Non-significantly different MM risk (non-different OR; *p* = 0.90). Non-significantly different CM risk (non-different OR; *p* = 0.94)
Nordeng et al., 2012 [67]	Retrospective cohort study	Antidepressants (SSRI, TCA, other)		Adjusted for depression (during pregnancy): Antidepressant: Non-significant risk for congenital M (aOR = 1.09; 95% CI 0.74–1.62), MM (aOR = 0.96; 95% CI 0.55–1.69) or cardiovascular M (aOR = 1.24; 95% CI 0.55–2.82) SSRI: Non-significant risk for congenital M (aOR = 1.22; 95% CI 0.81–1.84), MM (aOR = 1.07; 95% CI 0.60–1.91) or cardiovascular M (aOR = 1.51; 95% CI 0.67–3.43).
Reis et al., 2013 [53]	Retrospective cohort study	SSRI	Non-significant risk for MM (aOR = 1.05; 95% CI 0.94–1.17) Non-significant risk for CM (aOR = 0.99; 95% CI 0.82–1.21)	Not controlled
Ban et al., 2014 [32]	Population-based cohort study	SSRI	Non-significant risk for MM (aOR = 1.01; 95% CI 0.88–1.17) Non-significant risk for CM (aOR = 1.14; 95% CI 0.89–1.45) Non-significant risk for MM for single SSRI (paroxetine increased risk for CM (aOR = 1.78; 95% CI 1.09–2.88)	Compared to unmedicated depression (during pregnancy): Non-significant risk for MM (aOR = 0.93; 95% CI 0.78–1.11); non-significant risk for CM (aOR = 1.04; 95% CI 0.76–1.41). Non-significant risk for MM and CM for single SSRI.
Huybrechts et al., 2014 [56]	Population-based cohort study	SSRI	Significant CM risk (RR = 1.25; 95% CI 1.13–1.38)	Controlled for depression (lifetime): Non-significant CM risk (aRR = 1.06; 95% 0.93–1.22) Non-significant risk for single SSRI
Knudsen et al., 2014 [57]	Retrospective register-based cohort study	SSRI	Non-significant CM risk (aOR = 1.64; 95% CI 0.89–3.00). Significant risk for severe-CM (aOR = 4.03 1.75–9.26)	Not controlled
Furu et al., 2015 [37]	Retrospective population-based cohort study	SSRI	Significant risk for MM (aOR = 1.13; 95% CI 1.06–1.20); significant MM risk for fluoxetine (aOR = 1.25; 95% CI 1.10–1.42) and citalopram (aOR = 1.19; 95% CI 1.07–1.31). Significant CM risk (aOR = 1.15; 95% CI 1.05–1.26); significant CM risk for fluoxetine (aOR = 1.3; 95% CI 1.10–1.63). Sibling controlled: Non-significant risk for MM (aOR = 1.06; 95% CI 0.91–1.24) nor CM (aOR = 0.92; 95% CI 0.72–1.17)	Not controlled
Malm et al., 2015 [68]	Prospective register-based cohort study	SSRI (I trimester exposure)		Controlled for depression: Non-significant MM risk (aOR = 1.03; 95% CI 0.88–1.20)
Wemakor et al., 2015 [58]	Retrospective case–control study	SSRI	Significant risk for CM (aOR = 1.41; 95% CI 1.07–1.86) Significant severe CM risk (aOR = 1.56; 95 % CI 1.02–2.39), significant severe CM risk for sertraline (aOR = 2.88; 95% CI 1.09–7.61)	Not controlled
Nörby et al., 2016 [54]	Retrospective data linkage cross-sectional study	SSRI	Non-significant risk for total defects (OR = 1.0; 95% CI 0.9–1.2) and for CM (OR= OR = 1.0; 95% CI 0.9–1.2	Not controlled
Bérard et al., 2017 [69]	Register-based prospective cohort study	SSRI		Controlled for depression/anxiety (during pregnancy): Non-significant risk for MM (aOR = 1.07; 95% CI 0.93–1.22); Significant risk for citalopram for MM (aOR = 1.36; 95% CI 1.08–1.73). Significant risk for paroxetine for CM (aOR = 1.45; 95% CI 1.12–1.88)
Anderson et al., 2020 [59]	Retrospective population-based case–control study	SSRI (fluoxetine, citalopram, paroxetine, sertraline)	Significant risk for CM (aOR = 1.33; 95% CI 1.15–1.54) Non-significant for any neural tube defect (aOR = 0.99; 95% CI 0.74–1.33) Significant CM risk for fluoxetine (aOR = 1.52; 95% CI 1.14–2.02) and paroxetine (aOR = 1.52; 95% CI 1.04–2.23)	Controlled for exposure not in early pregnancy (1 month before conception + I trimester): Non-significant risk for CM (aOR = 1.14; 95% CI 0.87–1.51); non-significant for any neural tube defect (aOR = 0.81; 95% CI 0.48–1.36)
Melov et al., 2020 [60]	Retrospective cohort study	SSRI and SNRI		Controlled for history of any self-reported or medically diagnosed anxiety, depression, postpartum depression or bipolar disorder: Significant CM risk for SSRI/SNRI (aRR = 4.14; 95% CI 2.58–6.65) Significant CM risk for unexposed psychiatric disorder (aRR = 2.20; 95% CI 1.77–2.74)
Kolding et al., 2021 [64]	Retrospective register-based cohort study	SSRI (I trimester exposure)	Non-significant risk for severe CM (Prevalence Ratio = 1.09; 95% CI 0.52–2.30), non-significant risk for non-severe CM (PR = 1.38; 95% CI 1.00–1.92), significant CM risk for fluoxetine (PR= 2.69; 95% CI 1.46–4.97).	Controlled for depression/anxiety (not clear if during pregnancy): Non-significant severe CM risk (PR = 1.20; 95% CI 0.58–2.48), significant non-severe CM risk (PR = 1.50; 95% CI 1.02–2.20), significant non-severe CM risk for fluoxetine (PR = 2.40; 95% CI 1.24–4.63)
Rommel et al., 2022 [86]	Retrospective population-based cohort study	Antidepressants (SSRI, non-SSRI)		Compared to discontinuation of exposure and controlled for maternal psychiatric history: Antidepressant: Non-significant congenital M risk (aOR = 1.03; 95% CI 0.94–1.14), non-significant congenital M risk for I trimester exposure only (aOR = 1.10; 95% CI 0.94–1.30) SSRI: Non-significant congenital M risk (aOR = 1.01; 95% CI 0.91–1.12) Non-SSRI: Non-significant congenital M risk (aOR = 1.13; 95% CI 0.95–1.34)
Costei et al., 2002 [76]	Prospective cohort study	Paroxetine (III trimester exposure)	No CM cases	Not controlled
Schloemp et al., 2006 [70]	Prospective case–control study	Paroxetine	Non-significant risk for congenital M (RR = 0.76; 95% CI 0.18–2.53)	Not controlled
Cole et al., 2007 [93]	Retrospective cohort study	Paroxetine vs. other AD (SSRI, SNRI, SARI, TCAs, MAOIs) (I trimester exposure)	Significant risk for MM (aOR = 1.89; 95% CI 1.20–2.98). Non-significant risk for cardiovascular M (aOR = 1.46; 95% CI 0.74–2.88).	Not controlled
Diav-Citrin et al., 2008 [75]	Prospective cohort study	Paroxetine (median daily dose = 20 mg) (I trimester exposure)	Non-significant risk for cardiovascular M to paroxetine (aOR = 2.66; 95% CI 0.80–8.90).).	Not controlled
Einarson et al., 2008 [77]	Prospective cohort study	Paroxetine (I trimester exposure)	Non-significant risk for cardiovascular M (OR = 1.1; 95% CI 0.36–2.78)	Not controlled
Bakker et al., 2010 [78]	Retrospective population-based case–control study	Paroxetine (early exposure)	Non-significant risk for CM (aOR = 1.5; 95% CI 0.5–4.0). Significant risk for major atrium septum defects (OR = 5.7; 95% CI 1.4–23.7) Controlled vs. chromosomal or single gene disorder. OR not adjusted for smoking, teratogenic drug, maternal disease	Not controlled
Pastuszak et al., 1993 [71]	Prospective case–control study	Fluoxetine (I trimester exposure)	Non-significant risk (OR = 3.67; 95% CI 0.81–16.68) Non-significantly different MM rate (*p* = 0.3)	Not controlled
Chambers et al., 1996 [72]	Prospective cohort study	Fluoxetine (I trimester exposure)	Non-significant risk (OR = 1.35; 95% CI 0.5–3.3) Non-significantly different Major Structural Anomalies rate (*p* = 0.63)	Not controlled
Sivojelezova et al., 2005 [73]	Prospective cohort study	Citalopram (median daily dose = 0.345 mg/kg) (I trimester exposure)	Non-significant risk 1.09 (0.06–17.2) Non-significantly different MM rate (χ^2^, *p* = 0.52)	Not controlled
Diav-Citrin et al., 2008 [75]	Prospective controlled observational study	Fluoxetine (median daily dose = 20 mg) (I trimester exposure)	Significant risk for cardiovascular M (aOR = 4.47; 95% CI 1.31–15.27)	Not controlled
Klieger-Grossmann et al., 2012 [74]	Multicenter cohort study	Escitalopram	Non-significantly different MM rate (*p* = 0.83) Non-significant risk for MM (OR = 1.48; 95% CI 0.25–8.80)	Not controlled

MM = major malformations; CM = cardiac malformations; SSRI = selective serotonin reuptake inhibitors; SNRI = serotonin-noradrenaline reuptake inhibitors; AD = antidepressant; OR = odd ratio; aOR = adjusted OR; RR = relative risk; aRR = adjusted RR; CI = confidence interval.

**Table 3 ijerph-20-06565-t003:** SNRI use in pregnancy and risk of malformations.

Study	Design of the Study	Drug Class/Drug	Main Results (vs Not Exposed, General Population)	Main Results (vs Not Exposed, Psychiatric Population)
De Vries et al., 2021 [27]	Meta-analysis	SNRI (I trimester exposure)	All antidepressants significant CM risk (OR = 1.28; 95% CI 1.17–1.41), SNRI significant CM risk (OR = 1.69; 95% CI 1.37–2.10)	Not controlled
Lou et al., 2022 [30]	Meta-analysis	SNRI	Significant risk for CM (RR = 1.33; 95% CI 1.15–1.53) For I trimester exposure: non-significant MM risk (RR = 0.99; 95% CI 0.76–1.31), non-significant risk for congenital M (RR = 1.07; 95% CI 0.94–1.22)	Controlled for clinical indication: For I trimester exposure: non-significant MM risk (RR = 1.00; 95% CI 0.82–1.22), non-significant CM risk (RR = 1.17; 95% CI 0.95–1.42), non-significant congenital M risk (RR = 1.04; 95% CI 0.9–1.2)
Einarson et al., 2001 [81]	Prospective controlled study	Venlafaxine (daily dosage, mode = 75 mg)	Non-significant risk 1.6% vs. 0.7% (OR = 2.00; 95% CI 0.18–21.82) Non-significant MM risk (OR = 2.21; 95% CI 0.20–24.69)	Not controlled
Lennestål et al., 2007 [46]	Retrospective cohort study	SNRI, NRI	Non-significant congenital MM risk for SNRI/NRI (OR = 0.85; 95% CI 0.58–1.24)	Not controlled
Ramos et al., 2008 [31]	Retrospective case–control study	Antidepressants (I trimester exposure)		Controlled for depression/anxiety (during pregnancy): Non-significant MM risk (aOR = 0.94; 95% CI 0.51–1.75) Non-significant MM risk related to duration-of-exposure
Reis et al., 2010 [49]	Retrospective cohort study	SNRI (venlafaxine, mirtazapine)	Non-significant MM risk (OR = 1.00; 95% CI 0.73–1.37) non-significant CM risk (OR = 1.33; 95% CI 0.84–2.09)	Not controlled
Polen et al., 2013 [79]	Retrospective case–control study	Venlafaxine (early pregnancy)	Significant risk for some specific MM (OR = 2.31; 95% CI 1.30–4.08) Significant septal CM risk (aOR = 3.0; 95% CI 1.4–6.4) Significant anencephaly risk (aOR = 6.3; 95% CI 1.5–20.2) Significant LVOTO risk (aOR = 3.3; 95% CI 1.2–8.2) Significant cleft palate risk (aOR = 3.3; 95% CI 1.1–8.8)	Not controlled
Huybrechts et al., 2014 [80]	Population-based cohort study	SNRI	Significant CM risk (RR = 1.51; 95% CI 1.20–1.90)	Controlled for depression (lifetime): Non-significant CM risk (aRR = 1.20; 95% 0.91–1.57)
Furu et al., 2015 [37]	Retrospective cohort study	Venlafaxine (I trimester exposure)	Non-significant CM risk (aOR = 1.14; 95% CI 0.82–1.57)	Not controlled
Lassen et al., 2016 [82]	Review and meta-analysis of cohort studies	Venlafaxine Duloxetine (I trimester exposure)	Non-significant MM risk for venlafaxine (RR = 1.12; 95% CI 0.92–1.35) Non-significant MM risk for duloxetine (RR = 0.80; 95% CI 0.46–1.29)	Not controlled
Bérard et al., 2017 [69]	Register-based prospective cohort study	Venlafaxine		Controlled for depression/anxiety (during pregnancy): Non-significant MM risk (aOR = 1.10; 95% CI 0.87–1.38), Non-significant CM risk (aOR = 0.80; 95% CI 0.47–1.38)
Richardson et al., 2019 [83]	Prospective observational cohort study	Venlafaxine	Non-significant any MM risk (OR = 0.74; 95% CI 0.28–1.66), non-significant MM risk (OR = 1.05; 95% CI 0.26–3.19) Similar risk if restricted to I trimester exposure only	Not controlled
Anderson et al., 2020 [59]	Population-based, case–control	Venlafaxine (early exposure)	Significant risk for CM (aOR = 2.15; 95% CI 1.28, 3.60) Significant for any neural tube defect (aOR = 2.62; 95% CI 1.19, 5.75)	Controlled for exposure not in early pregnancy: Significant risk for CM (aOR = 1.94; 95% CI 1.09–3.47); significant risk for any neural tube defect (aOR = 2.58; 95% CI 1.01–6.57)
Kolding et al., 2021 [64]	Retrospective register-based cohort study	Venlafaxine (I trimester exposure)	Non-significant risk for severe CM (Prevalence Ratio = 2.13; 95% CI 0.89–5.13), Significant risk for non-severe CM (PR = 1.73; 95% CI 1.08–2.77)	Controlled for depression/anxiety (not clear if during pregnancy): Non-significant severe CM risk (PR = 1.78; 95% CI 0.67–4.70), significant non-severe CM risk (PR = 1.90; 95% CI 1.12–3.22)
Huybrechts et al., 2020 [80]	Retrospective nested cohort study	Duloxetine (I trimester exposure)	Significant MM risk (RR = 1.38; 95% CI 1.17–1.64)	Controlled for indication (depression, anxiety, pain): significant MM risk (aRR = 1.23; 95% CI 1.03–1.47) Controlled for all measured confounders (including indication and psychiatric comorbidities): Non-significant MM risk (aRR = 1.11; 95% CI 0.93–1.33) Non-significant cardiovascular M (aRR = 1.29; 95% CI 0.99–1.68) Controlled for measured and unmeasured confounders: Non-significant MM risk (aRR = 1.08; 95% CI 0.91–1.30)
Ankarfeldt et al., 2021 [84]	Retrospective cohort study	Duloxetine (I trimester exposure)	Non-significant risk for MM (OR = 1.12; 95% CI 0.87–1.43)	Controlled for comorbidities (anxiety, depression, affective disorder, severe stress reaction—not clear if during pregnancy): Non-significant MM risk (aOR = 0.98; 95% CI 0.74–1.30) Compared to discontinuers (exposure in the 356 days prior to LMP, no exposure during pregnancy): Non-significant MM risk (aOR = 0.80; 95% CI 0.56–1.14)

MM = major malformations; CM = cardiac malformations; SNRI = serotonin-noradrenaline reuptake inhibitors; AD = antidepressant; LMP = last menstrual period; LVOTO left ventricular outflow tract obstruction; OR = odd ratio; aOR = adjusted OR; RR = relative risk; aRR = adjusted RR; CI = confidence interval.

**Table 4 ijerph-20-06565-t004:** TCA use in pregnancy and risk of malformations.

Study	Design of the Study	Drug Class/Drug	Main Results (vs Not Exposed, General Population)	Main Results (vs Not Exposed, Psychiatric Population)
De Vries et al., 2021 [27]	Meta-analysis	TCAs (I trimester exposure)	Antidepressants significant CM risk (OR = 1.28; 95% CI 1.17–1.41) TCAs non-significant CM risk (OR = 1.02; 95% CI 0.82–1.25)	Not controlled
Simon et al., 2002 [65]	Retrospective cohort study	TCAs		Controlled for depression history (not clear if during pregnancy): Non-significant risk for MM (OR = 0.82; 95% CI 0.35–1.95), non-significant CM risk (OR = 0.50; 95% CI 0.05–5.53)
Källén et al., 2003 [41]	Retrospective case–control study	TCAs	Significant CM risk for TCA (OR = 1.14; 95% CI 1.07–2.91)	Not controlled
Davis et al., 2007 [44]	Register-based retrospective cohort study	TCAs (I trimester exposure)	Non-significant risk congenital MM (RR = 0.86; 95% CI 0.57–1.30) Non-significant risk for CM (RR = 0.92; 95% CI 0.23–3.70) 0.50 (0.12–2.01)	Not controlled
Ramos et al., 2008 [31]	Retrospective case–control study	TCAs (I trimester exposure)		Controlled for depression/anxiety (during pregnancy): Non-significant MM risk (aOR = 0.78; 95% CI 0.30–2.02) Non-significant duration-of-exposure MM risk
Reis et al., 2010 [49]	Retrospective cohort study	TCAs	Significant MM risk (OR = 1.36; 95% CI 1.07–1.72) Significant CM risk (OR = 1.63; 95% CI 1.12–2.36) Significant risk for MM 1.79 (1.52–2.11)	Not controlled
Ban et al., 2014 [32]	Population-based cohort study	TCAs (I trimester exposure)	Non-significant risk for MM (aOR = 1.09; 95% CI 0.87–1.38) Non-significant risk for CM (aOR = 1.03; 95% CI 0.65–1.63)	Compared to unmedicated depression (during pregnancy): Non-significant risk for MM (aOR = 1.02; 95% CI 0.79–1.32) Non-significant CM risk (aOR = 0.90; 95% CI 0.54–1.50)
Huybrechts et al., 2014 [56]	Retrospective population-based cohort study	TCAs	Non-significant CM risk (RR = 0.98; 95% CI 0.72–1.32)	Controlled for depression (lifetime): Non-significant CM risk (aRR = 0.77; 95% 0.52–1.14)
Bérard et al., 2017 [66]	Register-based prospective cohort study	TCAs		Controlled for depression/anxiety (during pregnancy): Non-significant risk for MM (aOR = 1.16; 95% CI 0.86–1.56)
Kolding et al., 2021 [64]	Retrospective register-based cohort study	TCAs (I trimester exposure)	Significant risk for non-severe CM (PR = 2.73; 95% CI 1.38–5.40)	Controlled for depression/anxiety (not clear if during pregnancy): Significant non-severe CM risk (PR = 3.36; 95% CI 1.68–6.71)

MM = major malformations; CM= cardiac malformations; OR = odd ratio; aOR = adjusted OR; RR = relative risk; aRR = adjusted RR; CI = confidence interval.

**Table 5 ijerph-20-06565-t005:** Atypical antidepressants use in pregnancy and risk of malformations.

Study	Design of the Study	Drug Class/Drug	Main Results (vs Not Exposed, General Population)	Main Results (vs Not Exposed, Psychiatric Population)
De Vries et al., 2021 [27]	Meta-analysis	Bupropion (I trimester exposure)	Antidepressants significant CM risk (OR = 1.28; 95% CI 1.17–1.41) Bupropione: Significant CM risk (OR = 1.23; 95% CI 1.01–1.49)	Not controlled No definition of indication
Einarson et al., 2003 [87]	Prospective controlled cohort study	Trazodone and nafazodone (I trimester exposure)	Non-significant risk for MM (OR = 0.50; 95% CI 0.09–2.72) Non-significantly different MM rates (*p* = 0.75)	Not controlled
Chun-Fai-Chan et al., 2005 [88]	Prospective cohort study	Bupropion	Non-significant risk for MM (OR = 0.19; 95% CI 0.00–4.09) Non-significantly different MM rates (*p* = 0.46)	Not controlled
Cole et al., 2007 [93]	Retrospective registry-based cohort study	Bupropion (I trimester exposure)	Non-significant congenital MM risk (aOR = 0.68; 95% CI 0.31–1.52), non-significant CM risk (aOR= 0.56; 95% CI 0.17–1.88) Non-significant congenital M risk (aOR= 0.69; 95%CI 0.36–1.34), non-significant CM (aOR= 0.54; 95% CI 0.19–1.51).	Controlled for exposure between 18 months before delivery and delivery (excluded I trimester) Controlled for exposure to another AD during I trimester
Huybrechts et al., 2014 [56]	Retrospective population-based cohort study	Bupropion	Non-significant CM risk (RR = 1.19; 95% CI 0.95–1.49)	Controlled for depression (lifetime): Non-significant CM risk (aRR = 0.92; 95% 0.69–1.22)
Alwan et al., 2010 [92]	Retrospective case–control study	Bupropion (early exposure)	Non-significant CM risk (aOR = 1.4; 95% CI 0.8–2.5), significant risk for left outflow CM (aOR = 2.6; 95% CI 1.2–5.7) Non-significant risk (OR = 1.12; 95% CI 0.67–1.87)	Not controlled No definition of indication
Anderson et al., 2020 [59]	Population-based, case–control	Bupropion (early exposure)	Non-significant risk for CM (aOR = 1.24; 95% CI 0.83–1.84)	Controlled for exposure 1 month before conception + I trimester: Non-significant risk for CM (aOR = 1.09; 95% CI 0.69–1.73)
Djulus et al., 2006 [89]	Prospective cohort study	Mirtazapine (mean daily dose = 30 ± 12 mg)	Non-significant risk for MM 0.02 (0.00–0.10) Non-significantly different MM rates (*p* = 0.69)	Not controlled
Reis et al., 2010 [49]	Retrospective cohort study	Mirtazapine + venlafaxine	Non-significant MM risk (OR = 1.00; 95% CI 0.73–1.37), non-significant CM risk (OR = 1.33; 95% CI 0.84–2.09)	Not controlled
Smit et al., 2015 [90]	Case series	Mirtazapine	No congenital malformations. Increased risk for PNAS 14/55 (25.9%)	Not controlled
Winterfeld et al., 2015 [85]	Prospective cohort study	Mirtazapine	Significant MM risk (OR = 3.27; 95% CI 1.04–10.25)	Controlled for exposure not in I trimester: non-significant MM risk (OR = 1.79; 95% CI 0.64–4.99)
Kolding et al., 2021 [64]	Retrospective register-based cohort study	Mirtazapine (I trimester exposure)	Significant risk for non-severe CM (PR = 3.04; 95% CI 1.16–7.97)	Controlled for depression/anxiety (not clear if during pregnancy): Significant non-severe CM risk (PR = 3.62; 95% CI 1.48–8.85)
Dao et al., 2023 [91]	Prospective observational cohort study	Trazodone vs. SSRI (early pregnancy)	Non-significant risk for MM (aOR = 0.20; 95% CI 0.03–1.77)	Not controlled

MM = major malformations; CM = cardiac malformations; AD = antidepressant; PNAS = poor neonatal adaptation syndrome. OR = odd ratio; aOR = adjusted OR; RR = relative risk; aRR = adjusted RR; CI = confidence interval.

**Table 6 ijerph-20-06565-t006:** Benzodiazepines use in pregnancy and risk of malformations.

Study	Design of the Study	Drug Class/Drug	Main Results (vs Not Exposed, General Population)	Main Results (vs Not Exposed, Psychiatric Population)
Dolovich et al., 1998 [94]	Meta-analysis (cohort and case–control studies)	BDZ (I trimester exposure)	Stratified for non-epileptic patients Cohort studies: non-significant risk for MM (OR = 0.90; 95% CI 0.61–1.35) Case–control studies: significant MM risk (OR = 3.01; 95% CI 1.32–6.84), significant oral cleft M risk (OR = 1.79; 95% CI 1.13–2.82)	Not controlled
Enato et al., 2011 [95]	Meta-analysis (cohort and case–control studies)	BDZ (I trimester exposure)	Cohort studies: non-significant risk for MM (OR = 1.06; 95% CI 0.91–1.25) Case–control studies: non-significant risk for CM (OR = 1.27; 95% CI 0.69–2.32)	Not controlled
Grigoriadis et al., 2019 [96]	Meta-analysis (cohort studies)	BDZ (I trimester exposure)	Non-significant risk for congenital MM (OR = 1.13; 95% CI 0.99–1.30); non-significant congenital MM risk for I trimester exposure (OR = 1.08; 95% CI 0.93–1.25) Non-significant risk for CM (OR = 1.27; 95% CI 0.98–1.65)	Controlled for psychiatric diagnoses: Non-significant congenital MM risk (OR = 1.03; 95% CI 0.79–1.32)
Grigoriadis et al., 2020 [105]	Meta-analysis (cohort studies)	HBRA	Non-significant congenital MM risk for I trimester exposure (OR = 0.87; 95% CI 0.56–1.36)	Controlled for psychiatric diagnosis: Non-significant congenital MM risk (OR = 0.88; 95% CI 0.44–1.77)
Ornoy et al., 1998 [106]	Retrospective cohort study	BDZ	Non-significant risk for MM (OR = 1.20; 95% CI 0.50–2.80)	Not controlled
Eros et al., 2002 [98]	Retrospective case–control study	BDZ (alprazolam, clonazepam, nitrazepam, medazepam, tofisopam)	Non-significant risk for congenital MM (OR = 1.3; 95% CI 0.9–1.8), non-significant risk for singular BDZ. Non-significant congenital MM risk for I trimester exposure (aOR = 1.4; 95% CI 0.9–2.3)	Controlled for psychiatric disease: Non-significant congenital MM risk (OR = 1.08; 0.9–3.9)
Wikner et al., 2007 [97]	Retrospective population-based study	BDZ or HBRA (early pregnancy)	Non-significant risk for congenital MM (OR = 1.10; 95% CI 0.90–1.35), Non-significant MM risk (OR = 1.22; 95% CI 0.97–1.52) Significant MM risk for BDZ (OR = 1.37; 95% CI 1.07–1.76)	Not controlled
Oberlander et al., 2008 [62]	Retrospective register-based cohort study	BDZ	Non-significant risk for MM 1.02 (0.72–1.44) Non-significant risk for CM 1.08 (0.45–2.60)	Controlled for any ICD-9 diagnosis containing depression (and dosage and duration of exposure): Non-significant MM Risk Difference (aRD = −0.41; 95% CI −1.51–0.69); Non-significant cardiovascular MM RD (aRD = −0.13; 95% CI −0.55–0.29).
Wang et al., 2010 [100]	Retrospective cohort study	Zolpidem	Non-significant risk for MM (aOR = 0.70; 95% CI 0.38–1.28)	Controlled for non I trimester exposure: Non-significant congenital MM risk (aOR = 0.74; 95% CI 0.38–1.44)
Wikner et al., 2011 [99]	Prospective population-based cohort study	HBRA	Non-significant congenital MM risk (OR = 0.89; 95% CI 95% CI 0.68–1.16) Non-significant risk for MM (OR = 0.95; 95% CI 0.69–1.30) Non-significant CM (aRR = 0.55; 95% CI 0.27–1.09)	Not controlled
Reis et al., 2013 [53]	Retrospective cohort study	BDZ or HBRA	BDZ: Non-significant risk for MM (aOR = 1.10; 95% CI 0.79–1.54), non-significant CM risk (aOR = 1.30; 95% CI 0.75–2.24) HBRA: Non-significant risk for MM (aRR = 0.86; 95% CI 0.57–1.10), non-significant CM risk (aRR = 0.26; 95% CI 0.63–0.94)	Not controlled
Ban et al., 2014 [102]	Retrospective cohort study	Diazepam Temazepam Zopiclone (I trimester exposure)	Non-significant MM risk for diazepam (aOR = 1.02; 95% CI 0.63–1.64), temazepam (aOR = 1.07; 95% CI 0.49–2.37), zopiclone (aOR = 0.96; 95% CI 0.42–2.20) or other anxiolytic/hypnotic drugs (aOR = 0.27; 95% CI 0.43–3.75) Non-significant CM risk for diazepam (aOR = 1.34; 95% CI 0.63–2.86), temazepam (aOR = 1.40; 95% CI 0.39–5.05) and zopiclone (aOR = 1.93; 95% CI 0.66–5.66)	Compared to unmedicated depression or anxiety: Non-significant congenital MM risk for diazepam (aOR = 0.99; 95% CI 0.61–1.61), temazepam (aOR = 1.04; 95% CI 0.47–2.32) and zoplicone (aOR = 0.93; 95% CI 0.40–2.15) Non-significant CM risk for diazepam (aOR = 1.29; 95% CI 0.60–2.80), temazepam (aOR = 1.31; 95% CI 0.35–4.92) and zoplicone (aOR = 2.03; 95% CI 0.69–6.02)
Tinker et al., 2019 [101]	Retrospective population-based case–control study	BDZ	Non-significant risk for any neural tube M (OR = 1.0; 95% CI 0.6–1.8), non-significant CM risk (OR = 0.9; 95% CI 0.6–1.2)	Not controlled
Noh et al., 2022 [103]	Retrospective population-based cohort study	BDZ (I trimester exposure)	Significant risk for CM (OR = 1.42; 95% CI 1.35–1.49) Non-significant congenital MM risk for <1 mg/day lorazepam-equivalent dose (aRR = 1.05; 95% CI0.99–1.12)	Controlled for psychiatric diagnosis (including sleep disorders): Significant congenital MM risk (aRR = 1.09; 95% CI 1.05–1.13) Significant CM risk (aRR = 1.15; 1.10–1.2)
Szpunar et al., 2022 [104]	Prospective cohort study	BDZ (I trimester exposure)		Controlled for psychiatric diagnosis (treated during pregnancy): Non-significant risk for MM (OR = 0.92; 95% CI 0.35–2.41)

MM = major malformations; CM = cardiac malformations; AD = antidepressant; HBRA = hypnotic benzodiazepine receptor agonists (Z drugs); OR = odd ratio; aOR = adjusted OR; RR = relative risk; aRR = adjusted RR; CI = confidence interval.

### 3.2. Complications in Pregnancy and Delivery

Complications in pregnancy and delivery, such as gestational hypertension, pre-eclampsia, gestational diabetes, intrauterine growth retardation, spontaneous abortion, preterm birth and postpartum hemorrhage associated with antenatal antidepressant or anxiolytic treatment, do not seem to overcome the baseline risk of the general population or the risks associated to the illness (Table 7).

Compared to the general population, a significant risk of *gestational hypertension* (OR 2.49, 95% CI 1.62–3.89) [107] and *pre-eclampsia* (OR 1.6, 95% CI 1.06–2.39) [108] in the SSRIs treated patients was reported in two [107,109] and one studies, respectively [108], while in other three studies [43,110], the risk was not significant. When considering any antidepressant use, significant gestational hypertension and pre-eclampsia risk was found in one [109] and three [49,80,108] studies, respectively. When antidepressant exposure was controlled for underlying condition (at least partially), a non-significant risk for SSRI was found in five studies for pre-eclampsia [80,107,111,112,113] and in two studies for gestational hypertension [68,109], while the risk remained significant for antidepressant use in three studies [108,109,111].

*Key findings*: The results are contradictory, but when controlled for depression in the mother, gestational hypertension or pre-eclampsia risk tended toward non-significant.

A possible risk of intrauterine *growth retardation* has been reported [114], but results are not clear, with another study reporting a non-significant small for gestational age risk [115]. Considering the *gestational diabetes* risk, exposure to antidepressants resulted in significant gestational diabetes risk in two studies [49,116] and a non-significant risk in one study [43]. A non-significant gestational diabetes risk was reported for any antidepressants [117,118] and for SSRI [116] when controlling for underlying depression in three studies overall.

*Key findings*: Gestational diabetes risk was reported mainly as non-significant.

A significant *spontaneous abortion* risk has been reported for women exposed to antidepressants compared to the general population in two meta-analyses [19,119] and seven observational studies [43,52,88,120,121,122,123], but another 14 studies did not confirm these results [46,49,54,75,84,87,89,117,124], in particular for SSRI use [38,71,72,73,81]. When an underlying psychiatric condition was considered, the results were not clear. Two studies, one of them a large population-based study, reported non-significant spontaneous abortion risk in women exposed to antidepressants when stratified for depression lifetime [84,122], but one study with a similar design reported significant spontaneous abortion risk [120].

*Key findings*: Spontaneous abortion or early pregnancy loss after antidepressant exposure are mainly reported as non-significant.

*Preterm birth risk* has been evaluated in 37 studies [39,40,43,46,49,52,54,67,68,71,72,73,80,86,87,89,115,125,126,127,128,129,130,131,132,133,134,135,136,137,138,139,140,141,142,143,144,145]; the risk of preterm birth after use of antidepressants in pregnancy was significant in four meta-analyses [131,132,134,140] and 16 studies [39,43,46,49,52,54,80,89,115,126,129,137,138,139,143,145] and non-significant in one meta-analysis [130] and 13 observational studies [40,42,68,71,73,87,125,128,130,133,135,136,144], in some cases, a non-significant risk for SSRI exposure only [39]. Interestingly, in five studies it was noted an increased risk of preterm birth exists for women with depression but without SSRI exposure during pregnancy [129,131,132,146,147]. When an underlying psychiatric condition was accounted for, the preterm birth risk was reported as significant in 11 studies [72,86,115,127,128,129,131,138,140,142,143] and non-significant in 9 studies [67,68,80,125,126,134,137,141,148]. A significant preterm birth risk was reported also after HBRA exposure, even after controlling for indication [149].

*Key findings:* Results for preterm birth risk are contradictory, even after controlling (at least partially) for an underlying condition.

The risk of *postpartum hemorrhage* after antidepressant exposure was reported as significant in four studies [49,80,150,151] and non-significant in another four studies [68,151,152,153] and seems to be very small to none [151]. After controlling (at least partially) for an underlying psychiatric condition, PPH risk was reported as non-significant in one study [68] and significant for SSRI [151] and duloxetine [80] in one and one study, respectively.

*Key findings*: Results are therefore contradictory and not conclusive.

**Table 7 ijerph-20-06565-t007:** Antidepressant use in pregnancy and risk of adverse maternal outcome.

Study	Design of the Study	Drug Class/Drug	Main Results (vs Not Exposed, General Population)	Main Results (vs Not Exposed, Psychiatric Population)
**Gestational Hypertension and Pre-eclampsia**				
Wen et al., 2006 [43]	Retrospective cohort study	SSRI (no escitalopram)	Non-significant pre-eclampsia risk (aOR = 1.20; 95% CI 0.90–1.61)	Not controlled
Toh et al., 2009 [107]	Retrospective cohort study	SSRI (exposure from 2 months before conception)	Significant any gestational hypertension risk (aOR 2.49, 95% CI 1.62–3.89), significant pre-eclampsia risk (aOR = 4.86; 95% CI 2.70–8.76), non-significant gestational hypertension without pre-eclampsia risk (aOR = 1.41; 95% CI 0.74–2.69)	Exposure during I trimester only: non-significant any GHT risk (aOR = 1.33; 95% CI 0.78–2.27) non-significant pre-eclampsia risk (aOR = 1.37; 95% CI 0.50–3.76) Non-significant GHT without pre-eclampsia risk (aOR = 1.30; 95% CI 0.69–2.46)
Reis et al., 2010 [49]	Retrospective cohort study	Antidepressants (SSRI, TCA, SNRI, MAOI) (early pregnancy)	Significant pre-eclampsia risk (aOR = 1.50; 95% CI 1.33–1.69), significant for early (aOR = 1.28; 95% CI 1.19–1.37) and late exposure (aOR = 1.38; 95% CI 1.25–1.53)	Not controlled
DeVera et al., 2012 [109]	Nested case–control study	Antidepressants (SSRI, TCA, SNRI, others)	Significant GHT risk for any antidepressants (OR = 1.52; 95% CI 1.10–2.09) SSRI (OR = 1.59; 95% CI 1.08–2.33) “other antidepressants” (OR = 3.89; 95% CI 1.39–10.94)	Adjusted for depression and anxiety history: Significant GHT risk for any antidepressant (aOR = 1.53; 95% CI 1.01–2.33) and for “other antidepressants” (aOR = 3.71; 95% CI 1.25–10.98); non-significant GHT risk for SSRI (aOR= 1.60; 95% CI 1.00–2.55), for TCA (aOR = 1.10; 95% CI 0.38–3.22) or for SNRI (aOR = 0.75; 95% CI 0.17–3.25); significant GHT risk for paroxetine (aOR = 1.81; 95% CI 1.02–3.23)
Palmsten et al., 2013 [111]	Retrospective cohort study	SSRI, SNRI, TCA (II and III trimester exposure)		Restricted to depression diagnosis: Significant pre-eclampsia risk for SNRI (RR = 1.52; 95% CI 1.26–1.83) and TCA monotherapy (RR = 1.62; 95% CI 1.23–2.12), non-significant risk for SSRI monotherapy (RR = 1.00; 95% CI 0.93–1.07)
Avalos et al., 2015 [108]	Retrospective cohort study	Antidepressants	Significant risk of pre-eclampsia for late exposure in depressed mothers (aRR = 1.70; 95% CI 1.30–2.23); Non-significant risk for late exposure in non-depressed mothers (aRR = 1.34; 95% CI 0.48–3.79) (95% CI overlapping)	Controlled for depression: significant pre-eclampsia risk for late exposure (aRR = 1.6; 95% CI 1.06–2.41) Late exposure and no depression compared to untreated depression: Non-significant pre-eclampsia risk (aRR = 1.29; 95% CI 0.44–3.79)
Malm et al., 2015 [68]	Prospective cohort study (n = 845,345)	SSRI	Non-significant hypertension risk (aOR = 1.09; 95% CI 0.98–1.20)	Controlled for depression: Non-significant hypertension risk (aOR = 1.10; 95% CI 0.97–1.26), significant for > 1 SSRI use (OR = 1.16; 95% CI 1.01–1.35)
Lupattelli et al., 2017 [112]	Retrospective registry-based cohort study	SSRI		Controlled for depression: Non-significant pre-eclampsia risk (aOR = 0.96; 95% CI 0.64–1.45), non-significant mild or severe pre-eclampsia risk
Huybrechts et al., 2020 [80]	Retrospective nested cohort study	Duloxetine	Significant pre-eclampsia risk for early exposure (OR = 1.67; 95% CI 1.44–1.93) and for late exposure (OR = 1.85; 95% CI 1.44–2.38)	Controlled for all measured confounders (including indication and psychiatric comorbidities): Non-significant pre-eclampsia risk for early exposure (aRR = 1.12; 95% CI 0.96–1.31) or late exposure (aRR = 1.04; 95% CI 0.80–1.35)
Vignato et al., 2023 [113]	Retrospective nested cohort study	SSRI (no fluvoxamine)		Controlled for depression: Non-significant pre-eclampsia risk (OR = 0.9; 95% CI 0.7–1.0; *p* = 0.05) Trend toward SSRI protective effect for pre-eclampsia risk.
**Gestational Diabetes**				
Wen et al., 2006 [43]	Retrospective cohort study	SSRI (no escitalopram)	Non-significant GD risk (aOR = 1.31; 95% CI 0.86–2.01)	Not controlled
Reis et al., 2010 [49]	Retrospective cohort study	Antidepressants (SSRI, TCA, SNRI, MAOI) (early pregnancy)	Significant GD risk (aOR = 1.37; 95% CI 1.08–1.75), significant GD risk for early exposure (aOR = 1.37; 95% CI 1.18–1.58)	Not controlled
Dandjinou et al., 2019 [116]	Retrospective case–control study	Antidepressants (SSRI, SNRI, TCA, other)		Controlled for depression: Significant GD risk for antidepressants (aOR = 1.19; 95% CI 1.08–1.30) Non-significant GD risk for history of depression and anxiety (aOR = 0.93; 95% CI 0.88–1.00) Significant risk for SNRI (aOR = 1.27; 95% CI 1.08–1.48) and TCA (aOR = 1.47; 95% CI 1.22–1.43) Non-significant risk for SSRI (aOR = 1.07; 95% CI 0.96–1.20) and others (aOR = 1.06; 95% CI 0.83–1.36)
Wartko et al., 2019 [117]	Retrospective cohort study	Antidepressants (SSRI, bupropion, venlafaxine)		Controlled for depression and anxiety: Non-significant GD risk (RR = 1.10; 95% CI 0.84–1.44)
Lupatelli et al., 2022 [118]	Retrospective cohort study	Antidepressants (I-II trimester exposure)		Restricted to mothers with depression and anxiety (during or in the 6 months before pregnancy): Non-significant GD risk for antidepressants (aRR = 0.88; 95% CI 0.43–1.82) or for low H1-receptor affinity antidepressants (aRR = 0.69; 95% CI 0.31–1.56)
**Intrauterine Growth Retardation**				
Grzeskowiak et al., 2012 [115]	Retrospective cohort study	SSRI	Non-significant SGA risk (aOR = 1.17; 95% CI 0.71–1.94)	Compared to psychiatric illness not exposed: Non-significant SGA risk (aOR = 1.13; 95% CI 0.65–1.94)
**Spontaneous Abortion and Stillbirth**				
Hemels et al., 2005 [119]	Meta-analysis	Antidepressants (SSRI, TCA, SNRI, DAA, MAOI)	Significant spontaneous abortion risk for antidepressants (RR = 1.45; 95% CI 1.19–1.77), SSRI alone (RR = 1.52; 95% CI 1.17–1.98) and DDA alone (RR = 1.65; 95% CI 1.02–2.69) Non-significant for TCA alone (RR = 1.23; 95% CI 0.84–1.78)	Not controlled
Rahimi et al., 2006 [19]	Meta-analysis	SSRI	Significant spontaneous abortion risk (OR = 1.70; 95% CI 1.28–2.24)	Not controlled
Pastuszak et al., 1993 [71]	Prospective case–control study	Fluoxetine (I trimester exposure)	Non-significantly miscarriage risk (OR = 1.9; 95% CI 0.92–3.92)	Not controlled
Chambers et al., 1996 [72]	Prospective cohort study	Fluoxetine (I trimester exposure)	Non-significantly different spontaneous abortion rate (*p* = 0.59) (OR = 1.80; 95% CI 0.96–3.36; calculated by Broy et al., 2010) and stillbirth rate (*p* = 1.00) Significantly increased therapeutic abortion rate (*p* = 0.002)	Not controlled
Kulin et al., 1998 [38]	Prospective cohort study	SSRI (fluvoxamine, paroxetine, sertraline)	Non-significantly different spontaneous abortion rate (*p* = 0.24) (OR = 1.53; 95% CI 0.85–2.74; calculated by Broy et al., 2010), therapeutic abortion rate (*p* = 0.30) or stillbirth rate (*p* = 0.50)	Not controlled
Einarson et al., 2001 [81]	Prospective cohort study	SSRI, venlafaxine	SSRI: non-significant spontaneous abortion risk (OR = 1.60; 95% CI 0.72–3.60; calculated by Broy et al., 2010) Venlafaxine: Non-significantly different spontaneous abortion rate (*p* = 0.24) (OR = 2.21; 95% CI 0.20–24.69) Non-significantly different therapeutic abortion rate (*p* = 0.18)	Not controlled
Einarson et al., 2003 [87]	Prospective cohort study	Trazodone and nafazodone (I trimester exposure)	Non-significantly different spontaneous abortion rate (*p* = 0.39) (OR = 1.82; 95% CI 0.85–3.88; calculated by Broy et al., 2010), therapeutic abortion rate (*p* = 0.06) or stillbirth rate (*p* = 0.89)	Not controlled
Chun-Fai-Chan et al., 2005 [88]	Prospective cohort study	Bupropion	Significantly increased spontaneous abortion rate (*p* = 0.009) (OR = 3.96; 95% CI 1.54–10.23; calculated by Broy et al., 2010) and therapeutic abortion rate (*p* = 0.015) Non-significantly different stillbirth rate (*p* = 0.9)	Not controlled
Sivojelezova et al., 2005 [73]	Prospective cohort study	Citalopram (III trimester exposure)	Non-significantly different live birth rate (*p* = 0.69) Non-significant spontaneous abortion risk for citalopram (OR = 1.10; 95% CI 0.50–2.45; calculated by Broy et al., 2010), Non-significant risk for any SSRI (OR = 1.07; 95% CI 0.53–2.14	Not controlled
Djulus et al., 2006 [89]	Prospective cohort study	Mirtazapine (mean daily dose = 30 ± 12 mg)	Non-significantly different spontaneous abortion rate (*p* = 0.12) (OR = 2.14; 95% CI 0.96–4.75; calculated by Broy et al., 2010), therapeutic abortion rate (*p* = 0.12) and stillbirth rate (*p* = 0.50)	Not controlled
Wen et al., 2006 [43]	Retrospective cohort study	SSRI (no escitalopram)	Significant fetal death risk (aOR = 2.23; 95% CI 1.01–4.93)	Not controlled
Lennestål et al., 2007 [46]	Retrospective cohort study	SSRI, SNRI, NRI	Non-significant intrauterine death risk for SSRI early exposure (OR = 0.8; 95% CI 0.5–1.2) and late exposure (OR = 1.2; 95% CI0.5–2.3) Non-significant intrauterine death risk for SNRI/NRI early exposure (OR = 1.7; 95% CI0.6–3.6) and late exposure (OR = 0.0; 95% CI 0.0–6.4)	Not controlled
Diav-Citrin et al., 2008 [75]	Prospective controlled observational study	Fluoxetine, paroxetine	Non-significant spontaneous abortion risk for fluoxetine (aOR = 1.27; 95% CI 0.76–2.13) or for paroxetine (aOR = 0.85; 95% CI 0.52–1.40)	Not controlled
Einarson et al., 2009 [123]	Retrospective cohort study	Antidepressants	Significant spontaneous abortion risk (RR = 1.63; 95% CI 1.24–2.14) and therapeutic abortion risk (RR = 3.25; 95% CI 1.48–7.14)	Not controlled
Nakhai-Pour et al., 2010 [120]	Nested case–control study	Antidepressants (SSRI, venlafaxine, TCA)		Controlled for depression, anxiety and bipolar disorder: Significant spontaneous abortion risk for any antidepressant (aOR = 1.68; 95% CI 1.38–2.06), for SSRI (aOR = 1.61; 95% CI 1.28–2.04) and for venlafaxine (aOR = 2.11; 95% CI 1.34–3.30); non-significant spontaneous abortion risk for TCAs (aOR = 1.27; 95% CI 0.85–1.91)
Colvin et al., 2012 [52]	Retrospective cohort study	SSRI	Significant stillbirth risk (OR = 1.07; 95% CI 0.72–1.58)	Not controlled
Kjaersgaard et al., 2013 [122]	Retrospective cohort study	Antidepressants (SSRI, TCAs, others)	Significant spontaneous abortion risk for all antidepressant exposure (aRR = 1.14; 95% CI 1.10–1.18) Significant SA risk for women without registered diagnosis of depression (aRR = 1.17; 95% CI 1.13–1.22),	Stratified for depression lifetime: Non-significant spontaneous abortion risk for antidepressants (aRR = 1.00; 95% CI 0.80–1.24)
Nörby et al., 2016 [54]	Retrospective data linkage cross-sectional study	Antidepressants (SSRI, SNRI, TCA, other)	Non-significant stillbirth risk (aOR = 1.2; 95% CI 1.0–1.5; *p* = 0.08); Significant perinatal death risk (OR = 1.3; 95% CI 1.1–1.5), non-significant for SSRI (OR = 1.2; 95% CI 1.0–1.5); Non-significant neonatal death risk (OR = 1.0; 95% CI 0.7–1.4)	Not controlled
Ankarfeldt et al., 2021 [84]	Retrospective cohort study	Duloxetine	Non-significant stillbirth risk (OR = 0.83; 95% CI 0.34–2.00)	Controlled for comorbidities (anxiety, depression, affective disorder, severe stress reaction—not clear if during pregnancy): Non-significant stillbirth risk (aOR = 1.18; 95% CI 0.43–3.19) Compared to discontinuers (exposure in the 356 days prior to LMP, no exposure during pregnancy): Non-significant stillbirth risk (aOR = 0.83; 95% CI 0.25–2.73)
**Postpartum Hemorrhage**				
Salkeld et al., 2008 [153]	Retrospective nested case–control study	SSRI (no escitalopram), non-SSRI (TCA, bupropion, venlafaxine)	Last 90 days of pregnancy exposure: Non-significant PPH for SSRI (aOR = 1.30; 95% CI 0.98–1.72) Non-significant PPH risk for non-SSRI (aOR = 1.12; 95% CI 0.62–2.01). Difference non-significant (*p* = 0.65) Secondary analysis: Last 180 days of pregnancy exposure: Significant PPH risk for SSRI (aOR = 1.32; 95% CI 1.03–1.70) Last 60 days of pregnancy exposure: Significant PPH risk for SSRI (aOR = 1.40; 95% CI 1.04–1.88)	Not controlled
Reis et al., 2010 [49]	Retrospective cohort study	Antidepressants (SSRI, TCA, SNRI, MAOI) (early pregnancy)	Significant bleeding during partus (aOR = 1.58; 95% CI 1.36–1.84), evidence for increased risk for late exposure. Significant after partus bleeding risk for early exposure only (aOR = 1.11; 95% CI 1.03–1.19)	Not controlled
Lindqvist et al., 2014 [150]	Prospective cohort study	SSRI	Significant PPH risk (aRR = 2.0; 95% CI 1.7–2.5), significant PPH risk for vaginal non-operative delivery (aRR = 2.3; 95% CI 1.8–3.0)	Not controlled
Malm et al., 2015 [68]	Prospective cohort study (n = 845,345)	SSRI	Non-significant bleeding risk (aOR = 1.07; 95% CI 0.95–1.21)	Controlled for depression: Non-significant bleeding risk (aOR = 0.83; 95% CI 0.71–0.96) Significant bleeding risk for psychiatric illness and no exposure (aOR = 1.29; 95% CI 1.13–1.48)
Huybrechts et al., 2020 [80]	Retrospective nested cohort study	Duloxetine	Significant postpartum hemorrhage risk (OR = 1.53; 95% CI 1.10–2.13)	Controlled for indication (depression, anxiety, pain): significant postpartum hemorrhage risk (aRR = 1.55; 95% CI 1.09–2.20) Controlled for all measured confounders (including indication and psychiatric comorbidities): Significant postpartum hemorrhage risk (aRR = 1.53; 95% CI 1.08–2.18), significant risk compared to SSRI (aRR = 1.48; 95% CI 1.03–2.12), non-significant compared to venlafaxine (aRR = 1.04; 95% CI 0.69–1.56) Controlled for measured and unmeasured confounders (hdPS adjustment): Significant postpartum hemorrhage risk (aRR = 1.48; 95% CI 1.04–2.10)
Skalkidou et al., 2020 [151]	Retrospective cohort study	SSRI	Significant PPH risk (aOR = 1.33; 95% CI 1.23–1.44)	Prior or current psychiatric illness without SSRI: Significant PPH risk (aOR = 1.07; 95% CI 1.02–1.12)
**Preterm Birth**				
Lattimore et al., 2005 [130]	Meta-analysis of prospective cohort studies	SSRI (III trimester exposure)	Non-significant prematurity risk (OR = 1.85; 95% CI 0.79–4.29)	Not controlled
Eke et al., 2016 [131]	Meta-analysis	SSRI	Significant PTB risk (aOR = 1.24; 95% CI 1.09–1.41)	Controlled for depression during pregnancy treated with psychotherapy alone: Significant PTB risk (OR = 1.17; 95% CI 1.10–1.25)
Chang et al., 2020 [132]	Meta-analysis of cohort studies	Antidepressants (SSRI, TCA, mirtazapine, venlafaxine)	Significant PTB risk for every pregnancy antidepressants exposure (aRR = 1.35; 95% CI 1.11–1.63) Non-significant PTB risk for every pregnancy SSRI exposure (aRR = 1.25; 95% CI 1.00–1.57). Significant PTB risk in depressed antidepressants exposed women (OR = 1.58; 95% CI 1.23–2.04) Significant PTB risk in depressed SSRI exposed women (OR = 1.46; 95% CI 1.32–1.61)	Not controlled
Vlenterie et al., 2021 [134]	Individual participant data meta-analysis	Antidepressants (SSRI, TCA, mirtazapine)	Significant preterm birth risk for antidepressants (aOR = 1.4; 95% CI 1.1–1.8) and for SSRI (aOR = 1.9; 95% CI 1.2–2.8)	Restricted to MDD or depressive symptoms: Non-significant preterm birth risk for antidepressants (aOR = 1.1; 95% CI 0.9–1.5) and for SSRI (aOR = 1.6; 95% CI 1.0–2.5)
Grigoriadis et al., 2022 [149]	Meta-analysis (cohort studies)	HBRA	Significant preterm birth risk (OR = 1.49; 95% CI 1.19–1.86), Significant I trimester exposure risk (OR = 1.42; 95% CI 1.09–1.86)	Controlled for psychiatric diagnosis: Significant preterm birth risk (OR = 1.44; 95% CI 1.25–1.67)
Kautzky et al., 2022 [140]	Meta-analysis	SSRI/SNRI	Significant preterm delivery pooled risk (OR = 1.75; 95% CI 1.18–2.52), significant lower gestational age (weeks) (Mean Difference= −0.47; 95% CI −0.74–−0.21)	Controlled for depression: Significant preterm delivery risk (OR = 2.36; 95% CI 1.35–4.15), non-significant lower gestational age (weeks) (MD = −0.36; 95% CI −0.81–0.8)
Pastuszak et al., 1993 [71]	Prospective case–control study	Fluoxetine (I trimester exposure)	Non-significantly different preterm birth rate (*p* = 0.22)	Not controlled
Chambers et al., 1996 [72]	Prospective cohort study	Fluoxetine (late exposure)		Compared to early exposed: Significant preterm birth risk (aRR = 4.8; 95% CI 1.1–20.8)
Ericson et al., 1999 [39]	Prospective cohort study	SSRI (citalopram, paroxetine, sertraline, fluoxetine), TCAs, other	Significant preterm birth risk (OR = 1.43; 95% CI 1.14–1.80) SSRI: non-significant preterm birth risk (OR = 1.30; 95% CI 0.81–2.10)	Not controlled
Casper et al., 2003 [141]	Prospective cohort study	SSRI		Compared to psychotherapy-only exposed major depressive disorder: Non-significantly different gestational age (mean age wk = 39.1 ± 1.1; *p* = 0.38), non-significantly increased preterm birth rate (*p* = 0.53)
Einarson et al., 2003 [87]	Prospective cohort study	Trazodone and nafazodone (I trimester exposure)	Non-significantly different age at birth (*p* = 0.57)	Not controlled
Hendrick et al., 2003 [40]	Prospective observational study	SSRI	Non-significant correlation for gestational age with medication (*p* = 0.42)	Not controlled
Sivojelezova et al., 2005 [73]	Prospective cohort study	Citalopram (III trimester exposure)	Non-significantly different preterm birth rate (*p* = 0.25)	Not controlled
Djulus et al., 2006 [89]	Prospective cohort study	Mirtazapine (mean daily dose = 30 ± 12 mg)	Significantly different preterm birth rate (*p* = 0.04), non-significantly different rate compared to another antidepressant exposure (*p* = 0.61)	Not controlled
Oberlander et al., 2006 [142]	Retrospective cohort data	SSRI		Compared to unexposed depression: Significantly lower gestational age (*p* < 0.001) and higher rates of preterm birth (*p* < 0.001) With propensity score matched for confounders: Non-significantly lower gestational age (*p* = 0.18) or higher rates of preterm birth (*p* = 0.61)
Wen et al., 2006 [43]	Retrospective cohort study	SSRI (no escitalopram)	Significant preterm birth risk (aOR = 1.57; 95% CI 1.28–1.92)	Not controlled
Ferreira et al., 2007 [135]	Retrospective cohort study	SSRI (III trimester exposure)	Non-significant prematurity risk (aOR = 2.4; 95% CI 0.9–6.3)	Not controlled
Lennestål et al., 2007 [46]	Retrospective cohort study	SSRI, SNRI, NRI	Significant preterm birth risk for SSRI (OR = 1.24; 95% CI 1.11–1.39) and SNRI/NRI (OR = 1.60; 95% CI 1.19–2.15)	Not controlled
Pearson et al., 2007 [133]	Retrospective cohort study	Antidepressants (SRI, TCA)	Non-significantly different gestational age (*p* = 0.68), non-significantly prematurity rate (*p* = 0.88)	Not controlled
Suri et al., 2007 [127]	Prospective cohort study	Antidepressants (SSRI - no fluvoxamine, nefazodone, venlafaxine, bupropione, nortriptyline)		Controlled for depression: Significant effect on gestational age (ANCOVA F = 6.0; df 2.87; *p* = 0.004), significant effect of dosage.
Maschi et al., 2008 [139]	Prospective cohort study	Antidepressants	Significant preterm delivery risk for chronic exposure (before and during entire pregnancy) (OR = 4.35; 95% CI 1.31–14.07)	Not controlled
Gavin et al., 2009 [125]	Retrospective cohort study	Antidepressants (+ other drugs)	Non-significant preterm delivery risk for exposure without depression (aOR = 1.4; 95% CI 0.8–2.4) or without depressive symptoms (CES-D) (aOR = 0.9: 95% CI 0.5–1.7) Non-significant preterm delivery for exposure with depression (aOR = 1.4; 95% CI 0.7–2.7) Significant medically indicated preterm delivery <35 week for exposure with depression (aOR = 3.6; 95% CI 1.1–12) Significant preterm delivery risk for exposure with depressive symptoms (CES-D) (aOR = 2.0; 95% CI 1.1–3.6) Non-significant crude OR for preterm delivery for depressive symptoms, depression and psychiatric medication use	Non-significant preterm delivery risk for depression not treated (aOR = 0.7; 95% CI 0.2–2.0) and for depressive symptoms not treated (aOR = 0.6; 95% CI 0.4–0.9)
Lund et al., 2009 [143]	Prospective cohort study	SSRI	Significant preterm birth risk (aOR = 2.02; 95% CI 1.29–3.16)	Compared to psychiatric condition history not exposed: Significant preterm birth risk (OR = 2.05; 95% CI 1.28–3.31)
Wisner et al., 2009 [126]	Prospective observational cohort study	SSRI	Significant preterm birth risk (aOR = 5.43; 95% CI 1.98–14.84), Non-significant risk for partial SSRI exposure (at least one trimester) (aOR = 0.86; 95% CI 0.11–6.92) Significant preterm birth risk for age < 31 years (aOR = 3.48; 95% CI 1.34–9.01)	MDD diagnosis and no exposure: Non-significant preterm birth risk (aOR = 3.71; 95% CI 0.98–14.13) Post-hoc: non-significant difference between SSRI exposure and MDD exposure without SSRI; both exposed groups significantly different from not exposed.
Lewis et al., 2010 [144]	Prospective case–control study	Antidepressants (SSRI, SNRI, NaSSA)	Non-significant preterm birth risk (OR = 4.52; 95% CI 0.47–43.41)	Not controlled
Reis et al., 2010 [49]	Retrospective cohort study	Antidepressants (SSRI, TCA, SNRI) (late exposure)	Significant preterm birth risk for SSRI (aOR = 1.46; 95% CI 1.31–1.63), TCA (aOR = 2.36; 95% CI 1.89–2.94) and SNRI (aOR = 1.98; 95% CI 1.49–2.63)	Not controlled
Latendresse and Ruiz, 2011 [145]	Prospective cohort study	SSRI (exposure before 20th week)	Significant preterm birth risk for SSRI exposure (OR = 11.7; 95% CI 2.2–60.7) and for high CRH level at 15–20 gestational week (OR = 6.6; 95% CI 1.4–31.5)	Not controlled
Colvin et al. 2012 [52]	Retrospective cohort study	SSRI	Significant preterm birth risk (aOR = 1.48; 95% CI 1.28–1.72)	Not controlled
El Marroun et al., 2012 [137]	Prospective cohort study	SSRI	Significant preterm birth risk (aOR = 2.14; 95% CI 1.08–4.25)	Group with significant maternal depressive symptoms not exposed to SSRI: Non-significant preterm birth risk (aOR = 1.10; 95% CI 0.77–1.59)
Grzeskowiak et al., 2012 [115]	Retrospective cohort study	SSRI	Significant preterm delivery risk (aOR = 2.46; 95% CI 1.75–3.50)	Compared to psychiatric illness not exposed: Significant preterm delivery risk (aOR = 2.68; 95% CI 1.83–3.93). Psychiatric illness not exposed: Non-significant preterm delivery risk (aOR = 0.94; 95% CI 0.79–1.12)
Nordeng et al., 2012 [67]	Retrospective cohort study	Antidepressants (SSRI, TCA, other)		Controlled for depressive symptoms (during pregnancy): Antidepressants: non-significant preterm birth risk (aOR = 1.21; 95% CI 0.87–1.69) SSRI: non-significant preterm birth risk (aOR = 1.28; 95% CI 0.90–1.84) Prior-only exposure: non-significant preterm birth risk (aOR = 1.12; 95% CI 0.84–1.49) Depressive symptoms at week 17: significant preterm birth risk (aOR = 1.13; 95% CI 1.03–1.25)
Yonkers et al., 2012 [148]	Prospective cohort study	SSRI (no fluvoxamine), venlafaxine, duloxetine	Non-significant preterm birth risk (aOR = 1.6; 95% CI 1.0–2.5)	Adjusted for psychiatric illness: Non-significant preterm birth risk for SSRI exposure without depression (aOR = 1.50; 95% CI 0.94–2.4), for SSRI exposure in depression (OR = 1.51; 95% CI 0.60–3.8) or depression not exposed to SSRI (OR = 0.86; 95% CI 0.44–1.7) Significant 34–37-week delivery risk for depressed exposed (aOR = 3.14; 95% CI 1.5–6.8) and for SSRI exposed not depressed (aOR = 1.93; 95% CI 1.2–3.2)
Sahingoz et al., 2014 [128]	Prospective case–control study	SSRI	Non-significant lower mean gestational age (*p* = 0.355)	Depressed not exposed: Significant lower mean gestational age (*p* = 0.033)
Malm et al., 2015 [68]	Prospective cohort study (n = 845,345)	SSRI	Non-significant preterm birth (aOR = 1.07; 95% CI 0.96–1.20)	Controlled for depression: Non-significant preterm birth risk (aOR = 0.84; 95% CI 0.74–0.96) Significant preterm birth risk for psychiatric illness and no exposure (OR = 1.27; 95% CI 1.13–1.44)
Nörby et al., 2016 [54]	Retrospective data linkage cross-sectional study	Antidepressants (SSRI, SNRI, TCA, other)	Significant 32–36 GA wk delivery risk (OR = 1.6; 95% CI 1.5–1.7), Non-significant < 32 gestational age week delivery risk (OR = 0.9; 95% CI 0.8–1.0) SSRI: Significant 32–36 GA wk delivery risk (OR = 1.6; 95% CI 1.4–1.7), Non-significant < 32 gestational age week delivery risk (OR = 0.8; 95% CI 0.7–1.0)	Not controlled
Viktorin et al., 2016 [129]	Retrospective cohort study	SSRI	Significant preterm birth risk (regression coefficient = 1.45; 95% CI 1.31–1.61)	Depression lifetime not exposed: Significant preterm birth risk (regression coefficient = 1.31; 95% CI 1.07–1.60)
Sujan et al., 2017 [138]	Retrospective cohort study	Antidepressants (mainly SSRI) (I trimester exposure)	Significant preterm birth risk (OR = 1.5; 95% CI 1.4–1.6)	Controlled for severe psychiatric illness history: Significant preterm birth risk (aOR = 1.35; 95% CI 1.28–1.42) SSRI only: Significant preterm birth risk (aOR = 1.27; 95% CI 1.20–1.35), even after restriction to siblings (OR = 1.33; 95% CI 1.16–1.53)
Huybrechts et al., 2020 [80]	Retrospective nested cohort study	Duloxetine	Significant preterm birth risk (RR = 1.33; 95% CI 1.22–1.46)	Controlled for all measured confounders (including indication and psychiatric comorbidities): non-significant preterm birth risk (aRR = 1.01; 95% CI 0.92–1.10) Controlled for measured and unmeasured confounders (hdPS adjustment): non-significant preterm birth risk (aRR = 0.99; 95% CI 0.90–1.09)
Rommel et al., 2022 [86]	Retrospective population-based cohort study	Antidepressant (SSRI, non-SSRI)		Compared to discontinuation of exposure and controlled for maternal psychiatric history: Significant risk for preterm birth (32–37 weeks) for: Antidepressants (aOR = 1.43; 95% CI 1.33–1.55—absolute risk difference = 2.1; 95% CI 1.7–2.6) SSRI (aOR = 1.28; 95% CI 1.18–1.39) Non-SSRI (aOR = 2.08; 95% CI 1.84–2.35) Significant very preterm birth risk (28–32 wk) for non-SSRI only (aOR = 1.86; 95% CI 1.33–2.59) Non-significant risk for extremely preterm birth (<28 wk)

GHT = gestational hypertension; DM = diabetes mellitus; PPH = postpartum hemorrhage; LBW = low birth weight; MDD = major depressive disorder; SSRI = selective serotonin reuptake inhibitors; SNRI = serotonin noradrenaline reuptake inhibitors; TCAs = tricyclic antidepressants; NaSSA = noradrenergic and specific serotonin antidepressants.

### 3.3. Neonatal Complications

#### 3.3.1. Persistent Pulmonary Hypertension of the Newborn

The association between the use of SSRIs in pregnancy and PPHN cannot be confirmed (Table 8). Two meta-analyses concluded for significant risk, especially for SSRI or SNRI exposure late in the gestational period [154,155], but the risk was non-significant for any-time exposure [155]., Risk has been evaluated in 11 studies and resulted significant in 7 [52,54,156,157,158,159,160] and non-significant in 3 studies [63,161,162]. However, the absolute risk is very low, ranging from 0.2% to 0.3% of the cases. In a large population-based cohort study of 29,822 live-born singleton children born to mothers who used antidepressants, the risk was not-significant when controlled for psychiatric history and antidepressant use before pregnancy (OR = 1.29; 95% CI 0.95–1.74) [156]. Also, three studies reported non-significant PPHN risk for antidepressant exposure after controlling for underlying conditions [86,156,159], while only one reported significant risk [54]. Key findings: the results remain not clear.

#### 3.3.2. Neonatal Abstinence Syndrome

Neonatal complications, such as neonatal abstinence syndrome, are the main consequence of antidepressants and anxiolytic therapy during pregnancy, with an incidence of 30% of the neonates or much more, depending on the epidemiological study; paroxetine is generally the SSRI most frequently implicated, followed by fluoxetine. Thirty-nine studies and one meta-analysis (Table 9) have been evaluated. The only meta-analysis evaluated reported a non-significant risk of poor adaptation symptoms in the newborn [130]. However, the risk of NAS after in utero use of antidepressants was significantly increased in 26 studies [42,44,46,49,54,68,73,76,115,135,140,142,143,163,164,165,166,167,168,169,170,171,172,173,174,175], and only 7 studies reported a non-significant risk for NAS symptoms or intensive care need [43,126,133,139,166,176,177]. It has been reported an incidence from 13.8% to 71.4%, depending on what signs and/or symptoms of NAS have been evaluated. Near-term exposure to antidepressants increased the risk of NAS, from 2-fold to 10-fold, depending on the comparison group. After controlling for an underlying condition in the mother, eight studies still reported significant NAS risk [54,68,72,86,115,140,143,173], while four reported non-significant risk for intensive care need [126,141,142,172]. The neonatal abstinence syndrome symptoms risk is significantly increased after antidepressant exposure during pregnancy.

It is more difficult to assess the incidence and severity of the neonatal withdrawal syndrome related to BDZ because, in the published studies, this potential adverse effect on the newborn has been assessed in pregnant women undergoing polypharmacological treatment (e.g., BDZ and antidepressants) [166,178] or exposed to other substances such as opiates. Overall, in these situations, neonatal withdrawal syndrome was observed in one out of three newborns. Concerning BDZ use in late pregnancy, the risk of NAS is as high as in the antidepressant group. The assessment of the NAS due to BDZ exposure during pregnancy is difficult, as most of the studies include other comedications (e.g., antidepressants) or they have collected opioid-exposed mothers. It has been reported that from 20% to 40% of the neonates with NAS were exposed to benzodiazepines.

Eventually, there is strong evidence of the association between opioid abuse with anxiety and depression, and the pharmacotherapy for substance abuse and addiction with methadone or buprenorphine treatment does not increase the risk of congenital malformations, while the risk of neonatal abstinence syndrome can overcome the 60% of the cases.

**Table 9 ijerph-20-06565-t009:** Neonatal abstinence syndrome/poor neonatal adaptation syndrome risk after antidepressant exposure during pregnancy.

Study	Design of the Study	Drug Class/Drug	Main Results (vs Not Exposed, General Population)	Main Results (vs Not Exposed, Psychiatric Population)
Lattimore et al., 2005 [130]	Meta-analysis of prospective cohort studies	SSRI (III trimester exposure)	Non-significant PNA risk (OR = 4.08; 95% CI 1.20–19.93; *p* = 0.069) Significant SCN/NICU admission risk (OR = 3.30; 95% CI 1.45–7.54)	Not controlled
Chambers et al., 1996 [72]	Prospective cohort study	Fluoxetine (late exposure)		Compared to early exposure: Significant risk of PNA (aRR = 8.7; 95% CI 2.9–26.6), Significant SCN admission (aRR = 2.6; 95% CI 1.1–6.9)
Costei et al., 2002 [76]	Prospective cohort study	Paroxetine (III trimester exposure)	12/55 infants exposed had prolonged hospitalization, significantly different complication prevalence (*p* = 0.03), significant respiratory distress risk (OR = 9.53; 95% CI 1.14–79.30)	Not controlled
Casper et al., 2003 [141]	Prospective cohort study	SSRI		Compared to psychotherapy-only exposed major depressive disorder: Non-significantly increased NICU admission rate (*p* = 0.06)
Laine et al., 2003 [165]	Prospective case–control study	SSRI (citalopram, fluoxetine) (late exposure)	Significant 4-fold increase (*p* = 0.008) in the serotoninergic symptoms score	Not controlled
Zeskind & Stephens, 2004 [174]	Case–control observational study	SSRI	Significantly increased rate of tremulousness (*p* = 0.038)	Not controlled
Källén et al., 2004 [163]	Retrospective cohort study	Antidepressants [SSRI, TCAs]	Risk for respiratory distress significant for all antidepressants (OR = 2.21; 95% CI 1.71–2.86), SSRI (OR = 1.97; 95% CI 1.38–2.83) and TCAs (OR = 2.20; 95% CI 1.44–3.35) Risk for hypoglycemia significant for all antidepressants (OR = 1.62; 95% CI 1.22–2.16) and TCAs (OR = 2.07; 95% CI 1.36–3.13); non-significant for SSRI (OR = 1.35; 95% CI 0.90–2.03) Risk for convulsions significant for all antidepressants (OR = 4.7; 95% CI 2.2–9.0) and for TCAs (OR = 6.8; 95% CI 2.2–16.0); non-significant for SSRI (OR = 3.6; 95% CI 1.0–9.3)	Not controlled
Oberlander et al., 2004 [166]	Prospective cohort study	SSRI (paroxetine, fluoxetine, sertraline) + clonazepam	30% of exposed infants showed PNA (likelihood ratio = 5.64; 95% CI 1.1–25.3), 25% in the SSRI-alone group (non-significant difference in incidence of symptoms between groups)	Not controlled
Sivojelezova et al., 2005 [73]	Prospective cohort study	Citalopram (III trimester exposure)	Significant NICU admission risk (RR = 4.2; 95% CI 1.71–10.26), non-significant risk for complications in general (RR = 1.5; 95% CI 1.0–2.4)	Not controlled
Levinson-Castiel et al., 2006 [170]	Prospective controlled cohort study	SSRI, venlafaxine	Of the 60 neonates exposed to SSRIs in utero, 18 developed NAS (30%; *p* < 0.001) Evidence of dose–response relationship for paroxetine	Not controlled
Wen et al., 2006 [43]	Retrospective cohort study	SSRI (no escitalopram)	Non-significant mechanical ventilation risk (aOR = 1.14; 95% CI 0.74–1.75), non-significant seizures risk (aOR = 3.87; 95% CI 1.00–14.99)	Not controlled
Davis et al., 2007 [44]	Retrospective case–control study	SSRI, TCA (III trimester exposure)	SSRI: significant risk of respiratory distress syndrome (RR = 1.97; 95% CI 1.65–2.35), metabolic disturbances (RR = 1.61; 95% CI 1.15–2.27), temperature regulation disorders (RR = 1.56; 95% CI 1.06–2.31) and convulsions (RR = 2.60; 95% CI 1.16–5.84) TCA: significant risk of respiratory distress syndrome (RR = 2.02; 95% CI 1.33–3.06), metabolic disturbances (RR = 2.15; 95% CI 1.04–4.44) and temperature regulation disorders (RR = 2.36; 95% CI 1.08–5.16)	Not controlled
Ferreira et al., 2007 [135]	Retrospective cohort study	SSRI, venlafaxine (III trimester exposure)	Significant neonatal behavioral signs risk for SSRI (aOR = 3.1; 95% CI 1.3–7.1) Non-significant admission to specialized care risk for SSRI (aOR = 2.4; 95% CI 0.8–6.9). Significantly increased prevalence for central nervous system signs (63.2%; *p* < 0.001), respiratory systems signs (40.8%; *p* < 0.001), tachycardia (16%; *p* = 0.006) and jaundice (22%; *p* = 0.001)	Not controlled
Lennestål et al., 2007 [46]	Retrospective cohort study	SSRI, SNRI, NRI	Significant respiratory problems risk for SSRI early exposure (OR = 1.17; 95% CI 1.03–1.23) and late exposure (OR = 1.72-; 95% CI1.41–2.11). Non-significant for SNRI/NRI Significant hypoglycemia risk for SSRI early exposure (OR = 1.17; 95% CI 1.02–1.33) and late exposure (OR = 1.32; 95% CI 1.05–1.68). Significant hypoglycemia risk for SNRI/NRI late exposure (RR = 2.11; 95% CI 1.01–3.89), non-significant for early exposure. Significant convulsions risk for SSRI late exposure (OR = 2.94; 95% CI 1.34–5.58). Non-significant for SNRI/NRI.	Not controlled
Pearson et al., 2007 [133]	Retrospective cohort study	Antidepressants (SRI, TCA)	Non-significantly different special care nursery admission rate (*p* = 0.084), significantly less days in SCN (*p* < 0.001)	Not controlled
Maschi et al., 2008 [139]	Prospective cohort study	Antidepressants	Non-significantly different rates of NAS, prolonged hospitalization or NICU admission	Not controlled
Boucher et al., 2008 [169]	Retrospective case–control study	Antidepressants (during last 3 weeks before delivery)	Increased risk of NAS symptoms (RR = 2.04; 95% CI 1.49–2.79) (aOR = 7.0; 95% CI 3.2–15.3)	Not controlled
Boucher et al., 2009 [175]	Prospective observational study	Venlafaxine (from II trimester)	7 neonates venlafaxine-exposed, 5 exhibited PNAS symptoms (71.4%). Evidence of dose–response relationship. Breastfeeding reduced symptoms in one case.	Not controlled
Lund et al., 2009 [143]	Prospective cohort study	SSRI	Significant NICU admission risk (aOR = 2.39; 95% CI 1.69–3.39)	Compared to psychiatric condition history not exposed: Significant NICU admission risk (OR = 2.04; 95% CI 1.42–2.94
Wisner et al., 2009 [126]	Prospective observational cohort study	SSRI	Non-significantly increased NICU admission rate for SSRI exposure or depression without SSRI exposure	Non-significant difference in NICU admission rate between SSRI exposure and depression without SSRI exposure
Galbally et al., 2009 [176]	Prospective case–control study	Antidepressants (SSRI, SNRI, NaSSA)	Non-significant respiratory distress risk (RR = 1.96; 95% CI 0.52–7.32), non-significant NICU admission risk (RR = 2.35; 95% CI 0.66–8.36), non-significant risk for neonatal hypoglycemia, hypothermia or convulsions. Significant risk for neonatal jaundice (RR = 4.67; 95% CI 1.11–19.94)	Not controlled
Warburton et al., 2010 [172]	Retrospective cohort study	SSRI + venlafaxine (in the 14 days before delivery)	Significantly higher respiratory problems rate (*p* < 0.001) and convulsions rate (*p* = 0.046) Non-significantly higher length of stay in hospital (*p* = 0.172) or jaundice rate (*p* = 0.095)	Controlled for depression severity: Non-significantly higher length of stay in hospital (*p* = 0.335), convulsion rate (same rate), jaundice rate (*p* = 0.568) or respiratory problems rate (*p* = 0.788)
Hale et al., 2010 [167]	Retrospective cross-sectional study	Antidepressants	Irritability 25%, significant problems with eating and sleeping 13%, jitteriness 10%, vomiting 10%, low body temperature 14%, shivering 5%, stiffness 5%, moaning 4%, convulsions 0.3%	Not controlled
Oberlander et al., 2006 [142]	Retrospective cohort data	SSRI	Significantly higher rates of hospital stay >3 days (*p* < 0.001) and preterm birth (*p* < 0.001), respiratory distress (*p* < 0.001) and jaundice (*p* = 0.01). Non-significantly different convulsions rate (*p* = 0.64).	Compared to unexposed depression: With propensity score matched for confounders: Non-significantly higher rated of hospital stay > 3 days (*p* = 0.07), jaundice (*p* = 0.45) or convulsions (*p* = 0.30) Significantly higher respiratory distress rate (*p* = 0.006)
Reis et al., 2010 [49]	Retrospective cohort study	SSRI, SNRI, TCA	Significant hypoglycemia risk (aOR = 1.56; 95% CI 1.36–1.79), significant for both early (aOR = 1.33; 95% CI 1.22–1.45) and late exposure (OR = 1.43; 95% CI 1.31–1.65) Significant respiratory diagnoses risk (aOR = 1.65; 95% CI 1.46–1.85), significant for both early (aOR = 1.34; 95% CI 1.25–1.44) and late exposure (OR = 1.62; 95% CI 1.47–1.79) Significant CNS diagnoses risk (aOR = 1.49; 95% CI 1.13–1.97), significant for both early (aOR = 1.31; 95% CI 1.11–1.56) and late exposure (OR = 1.50; 95% CI 1.19–1.88)	Not controlled
Udechuku et al., 2010 [179]	Systematic review	Antidepressant (SSRI, SNRI, TCAs, other)	Large database or registry studies: Higher risk of neonatal adaptation difficulties, 1.5 times risk of NICU admission after III trimester exposure and increased risks of respiratory distress and low APGAR scores. Evidence of increased risk of neonatal seizures. TCAs, only case report or case series: evidence of increased risk of adaptation difficulties. Venlafaxine, mirtazapine, duloxetine: evidence of increased NAS risk.	Not controlled
Altamura et al., 2012 [177]	Prospective cohort study	SSRI	Non-significantly different NICU admission rate	Not controlled
Colvin et al., 2012 [42]	Retrospective cohort study	SSRI	Significant “respiratory and cardio-vascular disorders specific to the perinatal period” risk (OR = 1.80; 95% CI 1.62–1.99) Significant respiratory distress of newborn (OR = 1.71; 95% CI 1.50–1.96) Significant NICU admission risk (aOR = 1.40; 95% CI 1.26–1.55)	Not controlled
Grzeskowiak et al., 2012 [115]	Retrospective cohort study	SSRI	Significant admission to hospital risk (aOR = 2.37; 95% CI 1.76–3.19)	Compared to psychiatric illness not exposed: Significant admission to hospital risk (aOR = 1.92; 95% CI 1.39–2.65) Psychiatric illness not exposed: Significant admission to hospital risk (aOR = 1.21; 95% CI 1.07–1.38)
Kieviet et al., 2013 [164]	Review	Antidepressant	Of all infants exposed to an SSRI in utero, 20%–77% develop symptoms of NAS, “most studies reported percentages around 30%”. “SSRI half-life seems to influence the risk of withdrawal and toxicity”. TCAs: “20–50% develops PNA” Venlafaxine PNA risk comparable to SSRI, mirtazapine one case report reported increased risk	Not controlled
Forsberg et al., 2014 [180]	Retrospective cohort study	Antidepressants (SSRI, SNRI)		Stratified for psychiatric illness during pregnancy (mainly depression): Neonatal characteristics of infants exposed: Hypoglycemia in 19%, Neonatal care admission in 13%, Respiratory diagnosis in 6%, Jaundice in 5%.
Malm et al., 2015 [68]	Prospective cohort study (n = 845,345)	SSRI	Significant breathing problems risk (aOR = 1.60; 95% CI 1.43–1.79) Significant NICU admission risk (aOR = 1.38; 95% CI 1.29–1.48)	Controlled for depression: Significant breathing problems risk (aOR = 1.40; 95% CI 1.20–1.62), trending toward increase for II-III trimester exposure (OR = 1.76; 95% CI 1.50–2.07) Significant NICU admission risk (aOR = 1.24; 95% CI 1.14–1.35), increased for II-III trimester exposure (OR = 1.51; 95% CI 1.37–1.66) Non-significant breathing problems risk for psychiatric illness and no exposure (aOR = 1.15; 95% CI 1.00–1.32) but significant NICU admission risk (aOR = 1.12; 95% CI 1.03–1.21)
Nörby et al., 2016 [54]	Retrospective data linkage cross-sectional study	Antidepressants (SSRI, SNRI, TCAs, mirtazapine/mianserine)	Significant NICU admission risk for SSRI (aOR = 1.5; 95% CI 1.4–1.5), higher for SSRI late exposure (aOR = 1.7; 95% CI 1.6–1.8). Significantly shorter NICU median duration of stay (*p* < 0.001). Significant risk for any-time SSRI exposure for any respiratory disorder (aOR = 1.6; 95% CI 1.5–1.7), hypoglycemia (aOR = 1.3; 95% CI 1.2–1.4) and CNS-related disorders (aOR = 1.5; 95% CI 1.2–1.8). Every risk increased with late SSRI exposure.	Late-exposure compared to early-exposure only: Significant NICU admission risk (aOR = 1.5; 95% CI 1.4–1.6), significant risk for SSRI (aOR = 1.4; 95% CI 1.2–1.6), for SNRI (aOR = 2.2; 95% CI 1.6–3.0) and for TCA (aOR = 1.7; 95% CI 1.1–2.7) Late compared to early SSRI exposure: Significant risk for any respiratory disorder (aOR = 1.9; 95% CI 1.6–2.2), hypoglycemia (aOR = 1.3; 95% CI 1.1–1.6) and CNS-related disorders (aOR = 1.8; 95% CI 1.1–2.9). Risk became non-significant in preterm infants.
Salisbury et al., 2016 [173]	Prospective cohort study	SSRI	Significantly increased stress–abstinence signs rate: Higher CNS signs (*p* = 0.003), higher hypotonia rate (*p* = 0.009), lower habituation scores (*p* = 0.017)	Compared to unexposed depression: Significantly higher CNS stress–abstinence signs (*p* < 0.008), hypotonia (*p* < 0.02)
Eleftheriou et al., 2017 [178]	Retrospective cross-sectional study	SSRI/SNRI and BDZ	13 newborns (50%) exposed in utero presented with NAS, 9 (69%) were exposed to SSRI/SNRI + BDZ and 4 (31%) were exposed only to SSRI/SNRI (*p* < 0.05). No indications of dose–response relationship (maternal drug daily dosage, neonate drug plasma level at birth)	Not controlled
Kautzky et al., 2022 [140]	Systematic review and meta-analysis (PRISMA)	SSRI or SNRI (III trimester exposure)	Significant risk for admission to NICU (OR = 1.74; 95% CI 1.43–2.11), convulsions (OR = 3.25; 95% CI 1.76–6.02), hypoglycemia (OR = 1.65; 95% CI 1.53–1.78), respiratory problems (OR = 1.96; 95% CI 1.80–2.14), temperature dysregulation (OR = 1.75; 95% CI 1.20–2.55), feeding problems (OR = 2.25; 95% CI 1.08–4.69)	Controlled for depression not exposed: Significant risk for admission to NICU (OR = 2.64; 95% CI 1.58–4.40), respiratory problems (OR = 2.85; 95% CI 1.26–6.43)
Wang & Cosci, 2021 [171]	Meta-analysis of observational studies	SSRI and venlafaxine (III trimester exposure)	Significant risk for: tachypnea (OR = 3.10; 95% CI 1.38–6.98), tremors (OR = 5.25; 95% CI 2.58–10.67), hypotonia (OR = 3.31; 95% CI 1.36–8.04), tachycardia (OR = 3.47; 95% CI 1.56–7.74), respiratory distress (OR = 3.87; 95% CI 1.38–10.89), hypertonia (OR = 6.86; 95% CI 1.18–40.08)	Not controlled
Shea et al., 2021 [168]	Prospective investigation study	SSRI/SNRI	13/80 (13.8%) infants presented mild NAS and 2% severe signs of NAS. Maternal CYP polymorphisms were non-significantly associated to NAS risk.	Not controlled
Rommel et al., 2022 [86]	Retrospective population-based cohort study	Antidepressants (SSRI, non-SSRI)		Compared to discontinuation of exposure and controlled for maternal psychiatric history: Significant PNAS risk for: antidepressants (aOR = 2.59; 95% CI 1.87–3.59—absolute risk difference = 0.5; 95% CI 0.4–0.6) SSRI (aOR = 2.51; 95% CI 1.79–3.50) non-SSRI (aOR = 3.05; 95% CI 1.91–4.87) Significant neonatal admission risk for: Antidepressants (aOR = 1.52; 95% CI 1.44–1.60—absolute risk difference = 6.3; 95% CI 6.2–6.4) SSRI (aOR = 1.43; 95% CI 1.35–1.51) Non-SSRI (aOR = 1.85; 95% CI 1.68–2.03)
Viguera et al., 2023 [181]	Retrospective registry-based cohort study	SSRI and SNRI vs. SGAs	“Overall, 129/384 (33.6%) infants presented with at least 1 sign of PNAS”, 66/191 (34.6%) in the SSRI/SNRI-exposed group. Most common signs were difficulties in breathing and feeding. (no statistical evaluation of between-group differences)	Not controlled

NAS = neonatal abstinence syndrome; PNA = poor neonatal adaptation; PNAS = postnatal adaptation syndrome; NSWS = neonatal SSRI withdrawal syndrome; NICU = neonatal intensive care unit; SCN = special care nursery; SSRI = selective serotonin reuptake inhibitors; SNRI = serotonin noradrenaline reuptake inhibitors; NaSSA= noradrenergic and specific serotonergic antidepressants; SGAs = second generation antipsychotics.

## 4. Current Guidelines for the Treatment of Depression in Pregnancy

Many countries have developed guideline recommendations for the treatment of depression during pregnancy (Table 10). All but one of these guidelines advise psychotherapy as initial treatment for mild to moderate depression. Only the American College of Obstetricians and Gynecologists guideline recommends antidepressants as the preferred initial therapy instead of psychotherapy. Three guidelines advise continuing antidepressants with the drug to which the patient has responded prior to pregnancy.

## 5. Discussion

The optimal choice for treating anxiety and depressive disorders during pregnancy should, as a first step, start considering the risks correlated with untreated illnesses. Many studies have found that a mental illness can affect a mother’s ability to obtain prenatal care and her ability to avoid unhealthy behavior. Moreover, these women have a poor quality of life, and they are more likely to smoke or use alcohol or other substances, which may worsen the pregnancy outcome. For these reasons, patient and health care provider preconception counseling can reduce maternal and neonatal risks by increasing maternal adherence to the antidepressant/anxiolytic treatment. In fact, while antidepressant medications have clear treatment benefits for the general population, in maternity, we must balance the risks of depression in pregnant women with the potential risks of medication passing through the placenta or breast milk to the fetus or newborn. Since ethical considerations preclude prospective, randomized controlled trials that would pose a danger to the fetus or newborn, physicians must rely on observational studies to make treatment decisions in this area.

Concerning untreated depression, many authors have associated it with adverse events in pregnancy, such as spontaneous abortion [189], bleeding during gestation [190], increased uterine artery resistance [191], gestational hypertension and subsequent pre-eclampsia [192,193], and also in the neonatal period as preterm deliveries [194], spontaneous early labor [195,196], low birth weight in babies [197], low Apgar scores [198], admission to a neonatal care unit [199], perinatal and birth complications and NAS-like symptoms [190,192,200] and neonatal growth retardation [201,202]. Furthermore, the discontinuation of therapy in the first trimester of pregnancy, or even the simple reduction of antidepressant therapy, entails a risk of exacerbation of the affective disorder in the former three months of gestation [203,204,205].

### 5.1. Congenital Malformations

**SSRI**. Selective serotonin reuptake inhibitors are the most common antidepressants used to treat anxiety and depressive disorders during pregnancy. All SSRIs cross the placenta. Most of the studies did not find an increased total rate of malformations associated with SSRIs as a grouped exposure [17,34,44,51,61,62,63]. Currently, the scientific data published on the relationship between the use of SSRIs and congenital anomalies suggest a low absolute risk. According to published studies and meta-analysis, sertraline appears to be the safest among SSRIs, while paroxetine seems to be the most reported drug associated with fetal heart malformations. Actually, cardiovascular anomalies are the main defect that could be significantly higher and the reason for growing concerns regarding the use of SSRIs, and paroxetine particularly, during pregnancy. Nevertheless, the current evidence of SSRI teratogenicity may be affected by various methodological weaknesses, lack of investigations using control groups of untreated depression, confounding by indication and recall bias [17,18,19,20,21,22,23,24,25,26,27,28,29,30,31,32,33,34,35,36,37,38,39,40,41,42,43,44,45,46,47,48,49,50,51,52,53,54,55,56,57,58,59,60,61,62,63,64,65,66,67,68,69,70,71,72,73,86,206].

**SNRI**. The existing literature on the safety of the SNRI medicines during pregnancy suggests that venlafaxine and duloxetine are relatively safe during pregnancy, in particular as far as major malformations are concerned. Bellantuono et al., in their review of 29 case-reports and case-series, suggested a lack of association between venlafaxine and an increased risk of major congenital malformations [207]. The same result was reported by Lassen et al., where the cumulative data of 3186 infants exposed to venlafaxine in pregnancy and 107 major malformations showed that venlafaxine was not associated with an increased risk of major congenital malformations [82].

Eventually, the position paper of the Canadian Pediatric Society in 2021 concludes that women who use an SSRI or SNRI during pregnancy should be counselled that the overall risk for congenital malformations or persistent pulmonary hypertension is low [208]. In conclusion, it remains difficult to determine whether SSRIs or SNRIs are associated with increased risk for teratogenic effects because, while some studies appear to demonstrate an association, others clearly do not. Importantly, however, when an association has been found, the absolute risk was low [69,79,80,81,82,83,84].

**Tricyclic antidepressants**. Although this class of antidepressants is much older than the SSRIs, there is a little experience about the potential teratogenicity. The risk of major congenital malformations seems to be not significant in all the epidemiological studies [32,44,65,133]. However, a slightly elevated risk of cardiac defects was identified on the basis of Swedish health databases, which include about 1600 pregnant women who were prescribed TCA in the first trimester [49]; in another register-based cohort study, the risk was not significant [69].

**Atypical antidepressants**. Almost all the epidemiological studies about the use of atypical antidepressants in pregnancy showed a non-significant risk for congenital malformations, but the number of patients evaluated till this year is very small, so it is difficult a clear assessment of the risk for this class of antidepressants [27,85,87,88,89,90,91,92,206].

**Benzodiazepines**. In benzodiazepines exposure, meta-analysis data from 11 cohort studies reported no association between fetal exposure to benzodiazepines and major malformations [66]. The same result was confirmed by an update to this meta-analysis by Enato et al., where the authors did not detect any statistically significant association between benzodiazepine exposure and major malformations [95]. To the same conclusions came a recent meta-analysis [96] and many other cohort studies [53,62,97,98,99,100,101,102,106]. Only one study reported an increased risk for cardiac malformations at the higher daily dose of BDZ during the first trimester of pregnancy [103].

### 5.2. Pregnancy Complications

Although the main doubt to treat or not treat women with depression is the risk of congenital malformations, there are other drug-induced maternal and fetal or neonatal complications that should be evaluated in this consensus. Gestational hypertension, pre-eclampsia, gestational diabetes, intrauterine growth retardation, spontaneous abortion, preterm birth and postpartum hemorrhage are topics of major concern.

#### 5.2.1. Antidepressants

Very few studies have focused on the potential obstetrical adverse effects of antidepressant use during pregnancy. Toh et al. reported that *gestational hypertension* was detected in 9.0% of the 5532 depressed women who were not treated with SSRIs and 19.1% of the 199 women who were treated with SSRIs [107]. Other authors have confirmed the significant risk of gestational hypertension in the SSRI-treated group [109]. On the other hand, Malm et al. found a non-significant risk of gestational hypertension in treatment with SSRI (4.5%) vs. untreated women with a psychiatric diagnosis (4.5%) [68]. It has been suggested that psychological conditions, such as stress, anxiety and depression [110,193], may trigger the pathogenic vascular processes that lead to this condition and need to be balanced against the effect induced by the pharmacological treatment.

Some authors have supposed a possible link between *pre-eclampsia* and the use of SSRIs in pregnancy [108,111]. Other authors did not observe such an association [112], while others found diametrically opposite results, observing that the use of SSRIs in pregnancy was associated with decreased pre-eclampsia after controlling for clinical confounders, such as depression severity, chronic hypertension, diabetes, body mass index and age [113].

*Intrauterine growth retardation* has been associated with several factors such as maternal medical conditions (antiphospholipid syndrome, gestational hypertension), tobacco and use of substances such as cocaine, alcohol and narcotics, maternal malnutrition, multiple gestation, infections, genetics and placental disorders. Also, the use of antineoplastic medications (e.g., cyclophosphamide), antiepileptic drugs (e.g., valproic acid) and antithrombotic drugs (e.g., warfarin) has been associated with an increased risk of fetal growth restriction. Concerning SSRI use in pregnancy, only a few studies have investigated a possible association with a possible intrauterine growth retardation and the results vary from a very low to a not-at-all risk [115].

In Europe, *gestational diabetes mellitus* is significant in pregnant women, around 11% of all pregnancies [209]. It has been postulated that all antidepressant exposure in pregnancy may be associated with a moderate risk of gestational diabetes [49], while other authors did not find any statistical evidence [43]. Eventually, recent publications on this topic indicated that there is no association between the use of SSRIs during pregnancy and the onset of gestational diabetes [116,117,118].

*Spontaneous abortion* is one of the most important adverse pregnancy outcomes, and the potential for maternal antidepressant use to influence the risk of spontaneous abortion has been studied by several authors. Broy et al., as well as Nakhai–Pour et al., reported a possible significant risk only for paroxetine and venlafaxine [120,121], while in a population-based study of more than one million pregnancies among women with a diagnosis of depression, the risk for spontaneous abortion after any antidepressant exposure was not significant [122].

*Preterm birth* is among the most complex and important challenges in obstetrics. It has been reported that in the United States, the preterm birth rate was 10.2% [210]. Many factors have been associated with preterm birth, including maternal demographics and characteristics, social and economic factors, medical complications, obstetric history and conditions specific to the current pregnancy. Also, the use of SSRIs has previously been found to be associated with an increased risk of preterm birth [43,49,115,126,127], while Lattimore et al., in their meta-analysis, showed a non-significant risk for preterm birth (1.85, 95% CI: 0.79–4.29) [130]. On the other hand, it is not clear whether this is attributable to the underlying depression or to SSRI use [125,126]. In fact, in the study of Vlenterie et al., women with depression were also observed to have an increased and significant risk of preterm birth (OR 2.2, 95% CI 1.7–3.0), highlighting the risks of untreated depression in pregnancy and suggesting low absolute risks of SSRI use during pregnancy [134]. Moreover, the heterogeneity of the studies was high, and no significantly increased risk of preterm birth was observed for SSRI use in the general pregnant female population [211].

About the *postpartum hemorrhage*, Corbetta–Rastelli et al., in their repeated cross-sectional study, analyzed delivery hospitalizations in the United States, and they reported that among an estimated number of 76.7 million delivery hospitalizations, 2.3 million (3.0%) were complicated by postpartum hemorrhage [152]. There are many causes of the maternal hemorrhage, and uterine atony is estimated to cause 70–80% of them. Moreover, a number of well-established risk factors, such as prolonged labor or chorioamnionitis, are associated with postpartum hemorrhage. The SSRI medications may interfere with normal platelet function and hemostasis by inhibiting serotonin reuptake in platelets, and they could be a risk factor as well. Several studies have observed a possible small increase in the risk of postpartum hemorrhage [150], while other studies reported no disproportionate risk of postpartum hemorrhage [153]. Studies reporting an increased risk of postpartum hemorrhage may be confounded by clinical management or other factors unrelated to pregnancy (i.e., treatment with NSAISs or a low dose of aspirin) or unrelated to the drug therapy (i.e., twin pregnancy). Lindqvist et al. did not exclude women with other bleeding disorders and cannot be certain that women taking SSRI continued to use the drug up to delivery [150], while Skalkidou et al. has identified an increased risk also for pregnant women with psychiatric illness but no medication exposure [151].

#### 5.2.2. Benzodiazepines

Concerning benzodiazepines and HBRA exposure during pregnancy, the most adverse outcomes assessed were spontaneous abortion, preterm birth delivery, low birth weight, low Apgar score at 5 min and neonatal intensive care unit admission. Grigoriadis et al., in their meta-analysis of fourteen studies, found that antenatal BZD exposure was significantly associated with an increased risk of all the above gestational complications [105], and other studies reported the same result [99,100]. However, other authors did not confirm any increased risk of low birth weight or small for gestational age [212,213].

### 5.3. Neonatal Complications

The risk for persistent pulmonary hypertension, neonatal adaptation difficulties of the newborn and long-term effects of in utero exposure are the main adverse effects in the offspring of pregnant women treated with antidepressants and/or anxiolytics.

#### 5.3.1. Persistent Pulmonary Hypertension

Persistent pulmonary hypertension has been associated with several conditions, while in the absence of cardiac or pulmonary disease, PPHN is referred to as idiopathic PPHN [214]. Concerning the association between the use of SSRI in pregnancy and PPHN, Chambers et al. were the first authors who reported an absolute risk of PPHN in SSRI-exposed infants of 1–2% [157], while the incidence in the general population has been reported as 1.9 per 1000 live births [215]. Subsequent studies have shown a much lower risk of PPHN, ranging from 0.2% to 0.3% absolute risk [154,155,158,216]. Other researchers could not confirm an association of SSRI exposure with PPHN [156]. Moreover, maternal depression, obesity, smoking and surgical delivery are also risk factors for PPHN, a fact that raises the difficulty of the interpretation of these data [161,162,217,218].

#### 5.3.2. Neonatal Abstinence Syndrome

Neonatal abstinence syndrome is a spectrum of clinical manifestations seen in neonates due to withdrawal from intrauterine psychotropic drug exposure. Neonatal symptoms, such as jitteriness, excitability, respiratory distress syndrome and, sporadically, seizures, were observed in newborns exposed in utero to antidepressants, benzodiazepines or opiates. Central nervous system hyperirritability leads to sleep disturbances and difficulty in breastfeeding. These symptoms are usually mild and transient [163,164]. The signs begin in the first days of life and continue in the typical infant for less than 2 weeks and for a month in very rare cases [165]. Paroxetine is generally the SSRI most frequently implicated, followed by fluoxetine [219]. The pathophysiology of this condition remains unknown. Due to different mechanisms of action, NAS from SSRI, TCA, BDZ and opioids may be attributable to a relative hyposerotoninergic state in the synaptic space, to a weak cholinergic transmission, to the reduced BDZ-induced GABAergic response, resulting in an increased glutamate excitatory activity and to an adenylyl cyclase super-activation [220] after tapering SSRI, TCA, BDZ and opioids, respectively. In addition to the receptor-specific differences between the SSRIs, the short half-life of paroxetine may account for more pronounced withdrawal symptoms. Inter-individual differences are also possibly caused by genetic polymorphism of the metabolizing enzymes and the 5–HT transporter activity [166,221]. When withdrawal signs are severe, with a Finnegan score above eight, pharmacotherapy should be initiated. Many authors reported a prevalence of NAS of 30–35% to more than 70% of the neonates exposed in utero to SSRIs [135,167,168,169,170,178,181,222,223]. A systematic review and meta-analysis of observational studies concerning NAS following late in utero exposure to SSRI reported that, depending on the symptoms of NAS, some manifestations could reach a high incidence, up to 77% of the SSRI-exposed newborns [171]. Eventually, other authors found that there were no differences seen between infants whose mothers reported discontinuation of the SSRI prior to the last month of pregnancy compared with infants whose mothers continued SSRI use through delivery [172,173].

Maternal use of benzodiazepines during late pregnancy has been associated with neonatal abstinence syndrome. The case reports of NAS in newborns exposed in utero to benzodiazepines have included descriptions of increased sedation, abnormal muscle tone, poor feeding, sleep problems, poor weight gain, tachypnea, irritability, loose stools, vomiting and vigorous sucking [97,136,148,224]. The incidence of NAS may vary from one to another in epidemiological studies; it has been reported that from 20% to 40% of the neonates with intrauterine exposure to benzodiazepines present symptoms with characteristics of neonatal abstinence syndrome [225,226]. The onset and duration of symptoms correlate with the pharmacokinetics and placental disposition of benzodiazepines [227]. In a large population-based retrospective cohort study, Sanlorenzo et al. reported an increased risk of developing NAS requiring pharmacological treatment in infants exposed antenatally to benzodiazepines [228]. For this reason, some authors suggest tapering the treatment of anxiolytics (or even antidepressants) before delivery in order to minimize the possibility of an appearance of NAS [97]; conversely, other studies conclude that reducing late third-trimester exposure to SSRIs would not substantially reduce the risk of the neonatal adverse outcomes [172].

About the pharmacologic characteristics of the BDZ (e.g., half-life) for a short-period treatment, the most used BDZ are the ones with a short half-life and no active metabolites like lorazepam. In the case of a BDZ with a long half-life and active metabolites like diazepam, it is expected an accumulation of the drug in adipose tissue and a prolonged transfer to the fetal compartment. On the other hand, the use of diazepam and other BDZs with long half-lives is less likely to be associated with NAS [229].

In addition, in newborns prenatally exposed to long-acting benzodiazepines or exposed to benzodiazepines shortly before delivery, it has been described a *floppy infant syndrome* (FIS) occurs. FIS symptoms described are hypotonia, inactivity, weak cry, lethargy, sucking difficulties, low Apgar score, hypothermia, apnea, cyanosis, hyperbilirubinemia, central nervous system depression that occur mainly within the first hours after labor and last up to 14 days [227].

### 5.4. Long-Term Effects

Concerning *the long-term impact* of these medications on exposed offspring, there are few data, and this possible effect is still poorly understood [230]. Given the important role of the serotonergic system throughout neurodevelopment and the presence of serotonin transport protein in early brain development, coupled with the ability of SSRIs to cross the placenta, it is reasonable to speculate that SSRIs in utero exposure, by increasing serotonin levels in the offspring, could have a long-term impact on the neurobehavioral outcome. Indeed, the serotonergic system regulates a range of brain development processes, including neuronal proliferation, migration, differentiation and synaptogenesis [231]. However, it should also not be forgotten that exposure to maternal stress-related disorders can also alter the developing serotonergic system in the fetus, as observed in mice [232].

Recent reviews reported the possible effects of perinatal SSRI exposure on social behavior, neurodevelopmental disorders and anxiety. Nevertheless, conflicting findings exist, and such evidence is not conclusive [231,233,234]. Population-based and cross-sectional studies showed that maternal depressive symptoms, rather than prenatal SSRI exposure, were related to higher levels of internalizing behavior [235,236] or higher levels of conduct and social problems in children [237]. In a similar study, maternal depressive symptoms without and with SSRIs were both associated with autistic traits in children [137]. However, in a comparative study, children prenatally exposed to venlafaxine, a selective serotonin and norepinephrine reuptake inhibitor, and an SSRI did not differ significantly from the children of mothers with untreated depression in any of the cognitive outcome or behavioral measures [238]. No data are available on other SNRI medications, such as duloxetine and vortioxetine or the atypical antidepressants. Limited and not updated data are available about tricyclic antidepressants. Indeed, these drugs have been largely replaced by SSRIs and SNRIs in depression management, mainly due to their adverse effects and lethality in overdose. Nonetheless, three studies investigating fetal exposure to TCA did not report any effects on the psychological and cognitive infant’s development [237,239]. In a large retrospective cohort study of 1,580,629 Swedish offspring using multiple statistical and methodical approaches to adjust for confounding factors, first–trimester antidepressant exposure was not significantly associated with the risk of being born small for gestational age or later autism spectrum disorder or attention-deficit/hyperactivity disorder [138].

Anxiety affects up to 15% of pregnant individuals worldwide and may require pharmacological treatment with anxiolytic agents, such as benzodiazepines. Z–hypnotics are benzodiazepine-related drugs mainly prescribed to treat mild insomnia. Children exposed to benzodiazepines and z-drugs in utero did not show significant neurodevelopmental effects [240]. A large retrospective cohort study suggests that previously described adverse neurodevelopmental outcomes associated with benzodiazepine exposure during pregnancy were likely to be accounted for by maternal genetic confounding [241]. However, due to limited data on the topic, some authors are still currently not certain whether or not prenatal exposure to BZDs and/or z–drugs might be associated with neurodevelopmental outcomes in the offspring [242].

### 5.5. Abuse Substances

There is strong evidence of the association between opioid abuse with anxiety and depression, as well as between depression and current tobacco or cannabis use [243,244,245]. This association is most commonly explained either by a causal relationship or a shared etiologic factor underlying both disorders. Some authors also found that each drug abused followed the onset of depression, except for LSD, which coincided with the onset of depression [246]. There is also a possible pathophysiologic link between the polygenic risk for major depressive disorder and opioid use disorder and opioid system activity, as evidenced by Love et al. [247].

The pharmacotherapy for substance abuse and addiction has been reviewed in several guidelines [248], and in National Institute for Health and Care Excellence [182] as well as in the Cochrane review of improving pregnancy outcomes [249], and reviews of psychosocial interventions and maintenance programs in pregnant women pregnancy exposed to opioids [250]. Treatment in pregnancy is based on a substantial body of evidence on the management of illicit opioid use in the general population, and this treatment has improved maternal and fetal outcomes compared to those seen with untreated opioid abuse [251]. The use of methadone in pregnancy was associated with better antenatal care, reduced maternal morbidity and better neonatal outcomes [252]. There are fewer data regarding buprenorphine in pregnancy, although evidence is increasing [253,254]. In a cohort study of 11,272 pregnant women exposed to buprenorphine and 5056 to methadone in late pregnancy, the risk of adverse maternal outcomes was similar among the two groups and similar to the baseline risk of the general population [255]. Zedler et al., in their meta-analysis, reported that buprenorphine was associated with approximately half the risk of preterm delivery and with greater birth weight (approximately 10 g) than the methadone treatment. No differences were observed for fetal death, congenital anomalies and fetal growth [251].

Regarding the neonatal abstinence syndrome, the risk is high as follows: a Cochrane review found no difference between methadone and buprenorphine in the incidence of neonatal abstinence syndrome requiring drug treatment [256]. In another study, 65.8% of the neonates born from mothers treated with methadone were diagnosed with neonatal opioid withdrawal syndrome requiring pharmacological interventions compared with 38.7% in the buprenorphine/naloxone group [257]. Lastly, the MOTHER study found that infants born to mothers treated with buprenorphine had shorter treatment for NAS, required fewer medications, and had shorter hospital stays [253].

Eventually, there is a possible link between childhood attention-deficit/hyperactivity disorder (ADHD) and substance abuse and depression [258], and pharmacological treatments for psychiatric disorders appear to mitigate the development of substance use disorders [259]. Furthermore, many authors suggest that long-term methylphenidate may reduce depression and suicide risk in ADHD patients [260]. During pregnancy, several studies reported that methylphenidate does not seem to increase the risk for major malformations [261,262], while Koren et al., in their meta-analysis, found a possible increased risk of cardiac malformations [263]. Despite these contrasting data, all authors agree that the discontinuation of methylphenidate therapy may be harmful with re-emergence of symptoms in ADHD patients [264].

### 5.6. Combined Treatment Options: Drug Treatment and Psychotherapy

We found limited evidence for the combination of drug treatment with psychotherapy in pregnant women. A meta-analysis (analyzing among others [65,142,143]) reported increased Preterm Birth risk in depressed women exposed to SSRI compared to psychotherapy alone [131]. Similar results are reported in a cohort study [141]. NICU admission rates do not seem to be increased with the combination treatment compared to psychotherapy alone [141].

## 6. Limitations of the Study

These consensus recommendations are comprehensive but not exhaustive. It is possible that we have omitted some interesting topics on which we did not query the expert panel. There is also the possibility of missed studies in the review strategy. Furthermore, some recommendations in this report are based on expert opinion, clinical experience and the extensive literature research, but we found a low number of high-quality studies with large samples. Eventually, the guidelines can be viewed as an expert consultation, to be weighed in conjunction with other information and in the context of each individual patient–physician relationship.

## 7. Conclusions

Several guidelines exist in other countries, regarding the treatment of anxiety and depressive disorders in pregnancy. Interestingly, based on the same available data, different guidelines advise different approaches. Our consensus is the first one in Italy, and our results draw a more optimistic behavior of the health practitioner in respect of the patient with depression. First and foremost, an adherence, meaning collaboration between the patient and the healthcare professional that respects the beliefs and wishes of a patient in determining whether, when and how medicines need to be taken should be achieved. Furthermore, this multidisciplinary consensus guide was developed to assist specialists and primary healthcare providers to risk stratifying and managing pregnant women with depression. It provides an overview of issues that may be of importance to healthcare providers involved in the management of anxiety and depressive disorders in pregnancy.

In conclusion, the recommendations engendered were the following:

***Recommendation 1.*** Qualified preconception counseling should be offered to all patients of reproductive age affected by anxiety and depressive disorders so that women and their partners can be more aware of the risks of untreated illness and of pharmacological treatment. Anxiety and depressive disorders are not a contraindication for gestation, nor it is a pharmacological treatment. *The Panel recommends that treatment should not be stopped abruptly in order to avoid relapse.*

***Recommendation 2.*** The Panel recommends continuing the medication to which the patient has responded well before pregnancy or switching to a safer drug if it can be equally effective (Table 11). When there is the need to start an antidepressant during pregnancy, drugs with a more favorable safety profile and more epidemiological data, such as the SSRI, should be preferred and prescribed at the lowest effective dose. Overall, most studies have not found an increased overall rate of malformations associated with SSRI exposure as a group. Moreover, studies that controlled for indication did not find an association with an increased rate of major or cardiac malformations. We can then conclude that it is the underlying condition and not the therapy indicated for it that is associated with an increased risk of MM or CM. Currently, the scientific data published on the relationship between the use of SSRIs and congenital anomalies suggest a low absolute risk. According to current studies and meta-analyses, sertraline appears to be the safest among SSRIs. *The Panel recommends continuing the medication to which the patient has responded before pregnancy or switching to a safer drug if it can be equally effective.*

***Recommendation 3.*** For the treatment of anxiety symptoms and short-term treatment of sleep disturbances, benzodiazepines can be administered during pregnancy. In order to minimize the pharmacological effects of the drug on the fetus, it is preferable to use drugs with a short elimination half-life, such as lorazepam, oxazepam and brotizolam.

The Panel concluded that benzodiazepines can be used during pregnancy and recommends using short half-life drugs.

***Recommendation 4.*** In opioid-dependent pregnant women, the use of methadone and buprenorphine is associated with better maternal and fetal outcomes compared with uncontrolled opioid misuse. There are no recommended pharmacological interventions for the abuse of psychoactive stimulants, cannabinoids or the new psychoactive substances in pregnancy. *The Panel recommends that opioid misuse treatment should not be stopped.*

***Recommendation 5.*** As for all drug use during pregnancy, possible fetal risk cannot be excluded; the risks are limited but not zero. Patients with depression who require standard treatments to control their disease must be followed up closely by an obstetrician once the pregnancy is confirmed (Table 11). The Panel recommends that cooperation between specialists (e.g., psychiatrists, obstetrics, toxicologist) should be encouraged. Whenever possible, every attempt should be made to offer multidisciplinary care to the patient, involving gynecologists and psychiatrists experienced in high-risk pregnancies. Coordination with the general practitioner is also recommended. *The Panel recommends offering multidisciplinary care when possible.*

***Recommendation 6.*** Neonatal abstinence syndrome is common, although the symptoms are mild and usually resolve quickly. The patient should be prepared to face this possible event with the utmost serenity (Table 11). The Panel does not agree with previous suggestions to stop or gradually taper the use of antidepressants before delivery in order to minimize the risk of NAS, as this may be associated with a predictable relapse of the precedent-treated depression during the postpartum. *The Panel recommends continuing pharmacological treatment till delivery.*

### 7.1. Ethical Considerations

When facing a woman in need of advice on the management of depression or anxiety disorders, consider informing her clearly and with informative materials (such as official leaflets or scientific papers). In many countries, there is no strict need to make an informed consent, but doing so can be useful to ensure that a shared decision is made in the most informed way. Regular follow-ups should be offered, and it is preferable to ensure a multidisciplinary approach. The evidence clearly suggests that treating underlying conditions during pregnancy is the most recommended choice. Drug treatment is mostly related to an enhanced risk of neonatal complications, but these complications usually resolve quickly and without sequelae. Women in need of advice should be informed clearly in order to ensure shared decision making. Regular multidisciplinary follow-up is recommended.

### 7.2. Future Research

More data are needed to clearly address indication bias. Most of the studies found do not control for underlying conditions such as depression or anxiety disorders. When this was considered, data were clearly in favor of treatment when considering malformation risk. More data are needed to clearly address the risk of complications for women during pregnancy and at the time of delivery. Limited data were found if not for spontaneous abortion and preterm birth risk, and even in these two cases, results are not clear even when indication bias was taken into account. Future research in the field should more clearly address long-term safety for children exposed to antidepressants or anxiolytics in utero, especially concerning neurodevelopmental outcomes.

## Figures and Tables

**Table 8 ijerph-20-06565-t008:** Antidepressant use in pregnancy and risk of persistent pulmonary hypertension of the newborn.

Study	Design of the Study	Drug Class/Drug	Main Results (vs Not Exposed, General Population)	Main Results (vs Not Exposed, Psychiatric Population)
Grigoriadis et al., 2014 [155]	Meta-analysis	SSRI	Non-significant PPHN risk for any time exposure (OR = 1.55; 95% CI 0.79–3.04) or for early exposure (OR = 1.23; 95% CI 0.58–2.60) Significant PPHN risk for late exposure (OR = 2.50; 95% CI 1.32–4.73)	Not controlled
Masarwa et al., 2019 [154]	Meta-analysis	SSRI, SNRI	Significant PPHN risk for SSRI or SNRI (aOR = 2.42; 95% CI 1.68–3.48) Safest treatment appeared to be sertraline (higher *p*; *p* = 0.83) (compared to fluoxetine OR = 0.34; 95% CI 0.11–0.96)	Not controlled
Chambers et al., 2006 [157]	Retrospective nested cohort study	Antidepressants (SSRI, TCA, bupropione, venlafaxine, trazodone) (II-III trimester exposure)	Significant PPHN risk for exposure after 20th week for antidepressant (aOR = 3.2; 95% CI 1.3–7.4) and for SSRI (aOR = 6.1; 95% CI 2.2–16.8) Similar risk if restricted to full-term birth	Not controlled
Källén et al., 2008 [160]	Retrospective observational study	SSRI	Significant PPHN risk for early exposure (RR= 2.38; 95% CI 1.19–4.25) and late exposure (RR = 3.57; 95% CI 1.16–8.33)	Not controlled
Andrade et al., 2009 [162]	Retrospective cohort study	Antidepressants (SSRI, TCA, other) (II-III trimester exposure)	Non-significant higher PPHN prevalence for antidepressants (prevalence ratio= 0.67; 95% CI 0.06–5.82) and SSRI (prevalence ratio = 0.79 95% CI 0.07–6.89)	Not controlled
Wichman et al., 2009 [63]	Retrospective cohort study	SSRI and venlafaxine (II-III trimester exposure)	Non-significantly different PPHN rate (*p* > 0.99)	Not controlled
Wilson et al., 2011 [161]	Retrospective case–control study	SSRI (II-III trimester exposure)	Non-significant PPHN risk (aOR = 0.0; 95% CI 0.0–3.0)	Not controlled
Colvin et al., 2012 [52]	Retrospective cohort study	SSRI	Significant PPHN risk (OR = 2.4; 95% CI 1.2–5.0)	Not controlled
Kieler et al., 2012 [158]	Retrospective cohort study	Antidepressants (SSRI, other with serotonin and norepinephrine activity)	Exposure in gestational week 20 or later: Significant PPHN risk for SSRI (aOR = 2.1; 95% CI 1.5–3.0) and for citalopram (aOR = 2.3; 95% CI 1.2–4.1), paroxetine (aOR = 2.8; 95% CI 1.2–6.7) and sertraline (aOR = 2.3; 95% CI 1.3–4.4). Non-significant for other antidepressants (OR = 2.9; 95% CI 0.9–8.9).	Not controlled
Huybrechts et al., 2015 [159]	Retrospective cohort study	Antidepressants (SSRI, non-SSRI) (II-III trimester exposure)	Significant PPHN risk for SSRI (OR = 1.51; 95% CI 1.35–1.69) and non-SSRI (OR = 1.40; 95% CI 1.12–1.75)	Restricted to depression: Non-significant PPHN risk for SSRI (aOR = 1.10; 95% CI 0.94–1.29) and non-SSRI (aOR = 1.02; 95% CI 0.77–1.35)
Nörby et al., 2016 [54]	Retrospective data linkage cross-sectional study	SSRI	Significant PPHN risk (OR = 1.3; 95% CI 1.0–1.7; *p* < 0.05)	Late compared to early exposure: significant PPHN risk (aOR = 1.7; 95% CI 1.1–2.8)
Munk-Olsen et al., 2021 [156]	Retrospective cohort study	Antidepressants (SSRI, SNRI, TCA, other)	Significant PPHN risk (OR = 1.99; 95% CI 1.58–2.51), significant for SSRI (OR = 1.68; 95% CI 1.28–2.21) and for non-SSRI (OR = 2.40; 95% CI 1.56–3.70) Significant PPHN risk for late exposure (OR = 2.54; 95% CI 1.59–4.05). Significant PPHN risk for late exposure for non-SSRI (OR = 3.83; 95% CI2.34–6.24), for SSRI (OR = 1.90; 95% CI 1.38–2.61) and for venlafaxine (OR = 2.24; 95% CI 1.06–4.73)	Controlled for psychiatric history and antidepressant use before pregnancy: Non-significant PPHN risk (aOR = 1.29; 95% CI 0.95–1.74), non-significant for SSRI (aOR = 1.16; 95% CI 0.83–1.61) and for venlafaxine (aOR = 1.15; 95% CI 0.56–2.36). Significant PPHN risk for non-SSRI exposure (aOR = 1.55; 95% CI 1.01–2.38) Significant PPHN risk for late exposure (aOR = 2.01; 95% CI 1.32–3.05) and for non-SSRI late exposure (aOR = 2.56; 95% CI 1.54–4.25)
Rommel et al., 2022 [86]	Retrospective population-based cohort study	Antidepressant (SSRI, non-SSRI)		Compared to exposure discontinuation and controlled for maternal psychiatric history: Non-significant risk for PPHN (aOR = 1.26; 95% CI 0.86–1.87 – absolute risk difference = 0.1; 95% CI 0.0–0.1) Non-significant for SSRI only and non-SSRI only

DM = diabetes mellitus; LBW = low birth weight; GA = gestational age; SGA = small for gestational age; SSRI = selective serotonin reuptake inhibitors; SNRI = serotonin noradrenaline reuptake inhibitors; TCAs = tricyclic antidepressants.

**Table 10 ijerph-20-06565-t010:** Guidelines for management of depression during pregnancy in other countries.

Scientific Society	Country	Main Recommendations
National Institute for Health and Care Excellence, 2014 [182]	UK	Advise psychotherapy and pharmacologic treatment for new-onset depression
German Society for Psychiatry and Psychotherapy, Psychosomatics and Neurology, 2017 [183]	Germany	Advise to continue antidepressants in pregnancy
Nordic Federation of Societies of Obstetrics and Gynecology, 2015 [184]	Norway	Advise to continue antidepressants in pregnancy and psychotherapy for new-onset depression
Dutch Society of Obstetrics and Gynaecology, 2012 [185]	Netherlands	No clear advice to continue antidepressants in pregnancy
Canadian Network for Mood and Anxiety Treatments, 2016 [186]	Canada	Advise psychotherapy and pharmacologic treatment for new-onset depression
Royal Australian and New Zealand College of Psychiatrists, 2015 [187]	Australia and New Zealand	Advise psychotherapy for new-onset depression
American College of Obstetricians and Gynaecologists, 2008 [188]	USA	Advise pharmacologic treatment for new-onset depression

**Table 11 ijerph-20-06565-t011:** Consensus panel recommendation. Colors are used to highlight risk of side effects for drug exposure both for pregnant women and for fetus.

Possible Complication to Address	Recommendation Based on Evidence of Risk for Pharmacological Treatment
Major Malformation or Cardiac Malformation	Treat underling condition, risk not increased. (Clear evidence)
Complications in Pregnancy or during Delivery	Treat underling condition with caution about potential adverse reaction. (Evidence not exaustive)
Neonatal Complications	Treat underling condition with close follow-up of the newborn. (Evidence of increased risk for PPHN and NAS symptoms)

## Data Availability

Not applicable.

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
