# Peer review of "Consensus Panel Recommendations for the Pharmacological Management of Pregnant Women with Depressive Disorders"

_ijerph, 2023, doi:10.3390/ijerph20166565_

Round 1

Reviewer 1 Report

  1. General Comments: a) The introduction provides a clear overview of the context and the need for a consensus on the pharmacological management of pregnant women with depressive disorders. b) The involvement of various scientific societies and the interdisciplinary team of experts is commendable. c) The identified areas of investigation are comprehensive and address crucial aspects related to the use of antidepressant and anxiolytic drugs during pregnancy. d) However, to enhance the quality and clarity of the paper, the following suggestions for major revisions are provided.
  2. Clarify the Methodology: a) Provide more details on the selection process for the interdisciplinary team of experts. Explain the criteria used to select individuals from each medical specialty and their roles in the consensus development. b) Elaborate on the "Nominal Group Technique" and how it was applied to synthesize the findings and arrive at the final consensus document. Clarify the number of systematic literature reviews conducted and their specific methodologies. c) Consider mentioning any potential conflicts of interest or funding sources related to the experts involved to ensure transparency and credibility.
  3. Strengthen the Results Section: a) Include a brief summary of the key findings within each area of investigation. This will help readers grasp the main conclusions before delving into the details. b) Provide more specific information about the literature evaluated, such as the number of studies included, their design (e.g., randomized controlled trials, cohort studies), and the quality assessment criteria employed. c) Consider presenting the findings in a tabular or graphical format to enhance readability and facilitate comparison between different aspects evaluated.
  4. Refine the Conclusion: a) The conclusion should summarize the main implications of the consensus findings and highlight the key recommendations for clinical practice. b) Consider emphasizing the need for ongoing research to address remaining gaps and uncertainties in the field, particularly regarding long-term outcomes for infants exposed to antidepressant/anxiolytic drugs in utero. c) If any specific recommendations have been developed as a result of the consensus, include them in the conclusion section and highlight their significance for healthcare providers.
  5. Overall Language and Clarity: a) Review the entire manuscript for grammar, sentence structure, and clarity of expression. Ensure that the information is presented in a concise and accessible manner. b) Avoid overly technical language when possible and provide explanations or definitions for specialized terms or acronyms to make the paper accessible to a wider audience. c) Maintain a consistent writing style throughout the document.
  6. Ethical Considerations: a) Discuss any ethical considerations related to the use of antidepressant and anxiolytic drugs during pregnancy, such as informed consent, monitoring of adverse effects, or potential risks to the fetus. b) If applicable, mention any ethical approvals or institutional review board (IRB) clearances obtained for the study.

Addressing these major revision suggestions will significantly enhance the clarity, rigor, and overall quality of the manuscript.

Overall Language and Clarity:

a) Review the entire manuscript for grammar, sentence structure, and clarity of expression. Ensure that the information is presented in a concise and accessible manner.

b) Avoid overly technical language when possible and provide explanations or definitions for specialized terms or acronyms to make the paper accessible to a wider audience.

c) Maintain a consistent writing style throughout the document.

Author Response

Reviewer 1

  1. a) Provide more details on the selection process for the interdisciplinary team of experts. Explain the criteria used to select individuals from each medical specialty and their roles in the consensus development.

We thank the reviewer for the recommendations. We have added the follow sentence:

All experts had a long-term experience and reviewed publications in the field. The role of the toxicologists was the evaluation of the risk of malformations after in utero use of the medicines; the interest of the gynecologists and the neonatologist was the complications in pregnancy and the neonatal abstinence syndrome respectively and the psychiatrists evaluated the risks due to the psychiatric disease of the patients involved in all published articles.   

  1. b) 1. Elaborate on the "Nominal Group Technique" and how it was applied to synthesize the findings and arrive at the final consensus document.

We have added the follow sentence:

“In our study, from September 2022 to June 2023, several NGT meetings were conducted in order to reach a consensus statement on the pharmacological management of pregnant women with depressive disorders.

The first face-to-face NGT was held in Bergamo, 8-10 September 2022, during the 33rd Conference of the European Network of the Teratology Information Services followed by adjustment of the first draft of consensus based of expert feedback via e-mail. Finally, after various online meetings, the research team reviewed all the materials and defined a list of consensus statements.”

  1. b) 2. Clarify the number of systematic literature reviews conducted and their specific methodologies.

We have added the follow sentence:

“The group of SSRI accounted in 17 meta-analysis and 51 cohort and case-control studies (32 retrospective, 17 prospective and 2 population-based cohort studies); the SNRI one was composed by 2 meta-analysis, 8 retrospective, 3 prospective and 2 population-based cohort studies. The tricyclic antidepressant and the atypical antidepressant group were composed by 1 meta-analysis, 7 retrospective, 1 prospective and 1 population-based cohort studies and 1 meta-analysis, 5 retrospective, 5 prospective and 1 population-based cohort studies respectively. Eventually, the BDZ accounted in 4 me-ta-analysis, 9 retrospective and 2 prospective cohort studies.”       

  1. c) Consider mentioning any potential conflicts of interest or funding sources related to the experts involved to ensure transparency and credibility.

We have added the follow sentence:

“None of the experts had any potential conflicts of interest or funding sources in order to ensure the experts’ transparency and credibility.”

Strengthen the Results Section:

  1. a) Include a brief summary of the key findings within each area of investigation. This will help readers grasp the main conclusions before delving into the details.

We have added a brief summary of the key findings within each area of investigation.

  1. b) Provide more specific information about the literature evaluated, such as the number of studies included, their design (e.g., randomized controlled trials, cohort studies), and the quality assessment criteria employed.

We have added the follow sentence:

“The group of SSRI accounted in 17 meta-analysis and 51 cohort and case-control studies (32 retrospective, 17 prospective and 2 population-based cohort studies); the SNRI one was composed by 2 meta-analysis, 8 retrospective, 3 prospective and 2 population-based cohort studies. The tricyclic antidepressant and the atypical antidepressant group were composed by 1 meta-analysis, 7 retrospective, 1 prospective and 1 population-based cohort studies and 1 meta-analysis, 5 retrospective, 5 prospective and 1 population-based cohort studies respectively. Eventually, the BDZ accounted in 4 me-ta-analysis, 9 retrospective and 2 prospective cohort studies.”       

  1. c) Consider presenting the findings in a tabular or graphical format to enhance readability and facilitate comparison between different aspects evaluated.

We have added a graphical format to enhance readability and facilitate comparison between different aspects evaluated.

Refine the Conclusion:

  1. a) The conclusion should summarize the main implications of the consensus findings and highlight the key recommendations for clinical practice.

We have added and highlight the key recommendations for clinical practice at the end of any recommendation.

  1. b) Consider emphasizing the need for ongoing research to address remaining gaps and uncertainties in the field, particularly regarding long-term outcomes for infants exposed to antidepressant/anxiolytic drugs in utero.

We have added a new paragraph about the future research:

More data are needed to clearly address indication bias. Most of the studies found do not control for underling condition as depression or anxiety disorders. When this was considered, data were clearly in favor of treatment when considering malformation risk.

More data are needed to clearly address risk of complications for women during pregnancy and at time of delivery. Limited data were found if not for Spontaneous Abortion and Preterm Birth risk, and even in these two cases results are not clear even when indication bias was taken in account.

Future research in the field should more clearly address long-term safety for children exposed to antidepressants or anxiolytic in utero, especially concerning neuro-developmental outcome.

  1. c) If any specific recommendations have been developed as a result of the consensus, include them in the conclusion section and highlight their significance for healthcare providers.

We have added the follow sentences in conclusion:

“Our Consensus is the first one in Italy and our results draw a more optimist behavior of the health practitioner in respect of the patient with depression. First and foremost, it should be achieved the adherence,  meaning a collaboration between the patient and healthcare professional that respects the beliefs and wishes of a patient in determining whether, when and how medicines need to be taken. Furthermore, this multidisciplinary consensus guide was developed to assist specialists and primary healthcare providers to risk stratify and manage pregnant women with depression. It provides an overview of issues that may be of importance to healthcare providers involved in the management of anxiety and depressive disorders in pregnancy.”

Overall Language and Clarity:

  1. a) Review the entire manuscript for grammar, sentence structure, and clarity of expression. Ensure that the information is presented in a concise and accessible manner.
  2. b) Avoid overly technical language when possible and provide explanations or definitions for specialized terms or acronyms to make the paper accessible to a wider audience.
  3. c) Maintain a consistent writing style throughout the document.

We have reviewed the entire manuscript for grammar, sentence structure, and clarity of expression, when possible, we avoid technical language and we tried to maintain a consistent writing style.

Ethical Considerations:

  1. a) Discuss any ethical considerations related to the use of antidepressant and anxiolytic drugs during pregnancy, such as informed consent, monitoring of adverse effects, or potential risks to the fetus.

We have added a new paragraph about ethical considerations:

When facing a woman in need for advice on management of depression or anxiety disorders, consider informing her clearly and with informative materials (such as official leaflets or scientific papers).

In many countries there is no strict need to make an informed consent but doing so can be useful to ensure that shared decision is made in the most informed way.

Regular follow-up should be offered and it is preferable to ensure a multidisciplinary approach.

Evidence clearly suggest that treating underling condition during pregnancy is the most recommended choice. Drug treatment is mostly related to enhanced risk of neonatal complications but these complications usually resolve quikly and without sequelae. Women in need of advice should be informed clearly in order to ensure shared decision making. Regular multidisciplinary follow-up is recommended.

  1. b) If applicable, mention any ethical approvals or institutional review board (IRB) clearances obtained for the study.

We thank the reviewer for the recommendation. In our review it is not applicable any ethical approval or institutional review board clearances.

The authors should include their inclusion and exclusion criteria in selection of the studies, and they also should include the PRISMA flow chart.

We have added the search strategy with the inclusion and exclusion of the studies and we have added the follow sentences:

“We have identified 620 records about the use of antidepressants during pregnancy and after exclusion of other languages other than English and experimental studies, we found 417 eligible studies. Searching only the full text articles we found 398 records. From them we excluded 61 papers concerning case-reports and case-series and 43 records were excluded for other reasons. From the remaining, 30 reports did not retrieved.”

The conclusions are consistent with the evidence and arguments presented and they address the main question posed. However, the authors should mention any change in their recommendation from the available guidelines recommendations.

We have added the follow sentence:

“Several  guidelines exist in other countries, regarding the treatment of anxiety and depressive disorders in pregnancy.  Interestingly, based on the same available data, different guidelines advise different approach. Our Consensus is the first one in Italy and our results draw a more optimist behavior of the health practitioner in respect of the patient with depression.”  

Reviewer 2 Report

The authors performed a review about the Consensus panel recommendations for the pharmacological management of pregnant women with depressive disorders. I would recommend the manuscript be accepted.

Even though most of the studies included in this review were not controlled and it covered several study types with varying degrees of strength of evidence. Overall, this concise review is well written and provides important evidence and recommendations for the pharmacological management of pregnant women with depressive disorders.

This topic is very important as the inconsistency across guidelines for management of depression during the perinatal period in the basic steps in the pathway to care strongly suggest the need for further research to solidify the evidence about the steps in the pathway, or at minimum build consensus around guidelines. As Lack of consensus across guidelines on specific issues relating to management of depression during the perinatal period. This is unfortunate given the high impact of lack of care during this period for both mothers and infants.

The authors should include their inclusion and exclusion criteria in selection of the studies, and they also should include the PRISMA flow chart.

The conclusions are consistent with the evidence and arguments presented and they address the main question posed. However, the authors should mention any change in their recommendation from the available guidelines recommendations.

The author should include references regarding the adopted guidelines for management of perinatal depression.

The tables are informative but a table about the available perinatal depression guidelines and recommendations must be included.

only minor grammatical review required

Author Response

Reviewer 2

The author should include references regarding the adopted guidelines for management of perinatal depression. The tables are informative but a table about the available perinatal depression guidelines and recommendations must be included.

We thank the reviewer for the recommendations. We have added the table (Table 10. Guidelines for management of depression during pregnancy in other countries) and the references regarding the adopted guidelines for management of perinatal depression.

Reviewer 3 Report

The review ‘Consensus Panel Recommendations for the Pharmacological Management of Pregnant Women with Depressive Disorders’ is highly topical and well written manuscript.

At the section ‘4.2.1. Antidepressants’ sub-section ‘About the post–partum hemorrhage’ specify more details about the studies that show the risk of SSRI on of postpartum hemorrhage [125].

Add the limitations of this study or limitations on the Recommendations. Taking into account that there is missing discussion or recommendation on efficacy of the combined treatment option: pharmacological + psychotherapy.

Author Response

Reviewer 3

At the section ‘4.2.1. Antidepressants’ sub-section ‘About the post–partum hemorrhage’ specify more details about the studies that show the risk of SSRI on of postpartum hemorrhage [125].

We thank the reviewer for the recommendations. We have added the follow sentence:

Studies reporting an increased risk of post–partum hemorrhage may be confounded by clinical management or other factors, unrelated to pregnancy (i.e., treatment with NSAISs or low-dose of aspirin) or unrelated to the drug therapy (i.e., twin pregnancy). Lindqvist et al did not exclude women with other bleeding disorders and cannot be certain that women taking SSRI continued to use the drug up to delivery [125], while Skalkidou et al have identified an increased risk also for pregnant women with psychiatric illness but no medication exposure [126].

Add the limitations of this study or limitations on the Recommendations.

We have added the follow paragraph:

“Limitations of the study

These guidelines are comprehensive but not exhaustive; possible, we omit some interesting topics on which we did not query the expert panel and there is also the possibility of missed studies in the search strategy. Furthermore, some recommendations in this report are based on expert opinion, experience and the literature research but we found a low number of high quality studies with large samples. Eventually,  the guidelines can be viewed as an expert consultation, to be weighed in conjunction with other information and in the context of each individual patient-physician relationship.”

Taking into account that there is missing discussion or recommendation on efficacy of the combined treatment option: pharmacological + psychotherapy.

We have added the follow paragraph:

4.6. Combined treatment options: Drug treatment and psychotherapy

We found limited evidence for the combination of drug treatment with psycho-therapy in pregnant women. A meta-analysis (analyzing among others [64], [240] and [241]) reported increased Preterm Birth risk in depressed women exposed to SSRI com-pared to psychotherapy alone [130]. Similar results are reported in a cohort study [239]. NICU admission rates don’t seem to be increased with the combination treatment com-pared to psychotherapy alone [239].
